# Polar Mesospheric Summer Echo (PMSE) Multilayer Properties During Solar Maximum and Solar Minimum

Dorota Jozwicki[1], Puneet Sharma[2], Devin Huyghebaert[1], and Ingrid Mann[1]

[1]Department of Physics and Technology, UiT the Arctic University of Norway, 9019 Tromso, Norway
[2]Department of Automation and Process Engineering, UiT the Arctic University of Norway, 9019 Tromso, Norway

**Correspondence:** Dorota Jozwicki (dorota.s.jozwicki@uit.no)

**Abstract.**

Polar Mesospheric Summer Echoes (PMSE) are radar echoes that are measured in the upper atmosphere during the summer months and that can occur in several layers. In this study, we aimed to investigate the relationship between PMSE layers ranging from 80 to 90 km altitude, and the solar cycle. We investigated 230 hours of observations from the EISCAT VHF radar located near Tromsø, Norway, from the years 2013, 2014 and 2015, during solar maximum and the years 2019 and 2020 during solar minimum and applied a previously developed classification model to identify PMSE layers. Our analysis focused on parameters such as the altitude, thickness, and echo power in the PMSE layers, as well as the number of layers present. Our results indicate that the average altitude of PMSE, the echo power in the PMSE and the thickness of the layers is on average higher during solar maximum than during solar minimum. In the considered observations, the electron density at 92 km altitude and the echo power in the PMSE are positively correlated with the thickness of the layers, except for four multi layers at solar minimum. We infer that higher electron densities at ionospheric altitudes might be necessary to observe multi-layered PMSE. We observe that the thickness decreases as the number of multi-layers increases. We compare our results with previous studies and find that similar results regarding layer altitudes were found in earlier studies using observations with other VHF radars. We also observed that the bottom layer in in the different sets of multilayers almost always aligned with the Noctilucent Clouds (NLC) altitude reported by previous studies of 83.3 km altitude. Also, an interesting parallel is seen between the thickness of NLC multi layers and PMSE multi layers, where both NLC and PMSE have a similar distribution of layers greater than 1 km in thickness. Future studies that include observations over longer periods would make it possible to distinguish the influence of the solar cycle from possible other long-term trends.

## 1 Introduction

During the summer months, radars can measure a phenomenon in the upper atmosphere called Polar Mesospheric Summer Echoes (PMSE). PMSE are strong radar echoes that typically form at heights between 80 and 90 kilometers and in regions of extremely cold temperatures. They are observed at mid and high latitudes and their height and thickness varies over time, (Rapp and Lübken, 2004). Figure 1 shows a typical example of a PMSE event where these variations can be seen. The PMSE formation is linked to the presence of turbulence, free electrons, and charged aerosols. The charged aerosols contain water

ice, which requires the presence of low temperatures, sufficient water vapor, and nucleation centers to foster heterogeneous condensation, (Latteck et al., 2021), (Cho and Röttger, 1997), (Rapp and Lübken, 2004). The mesopause, which marks the boundary between the mesosphere and the thermosphere, is characterized by the lowest temperatures in the atmosphere. Such low temperatures at PMSE altitudes are conducive to ice formation. Meteor Smoke Particles (MSP), produced by meteor ablation and recondensation have been proposed as potential condensation nuclei along with several other potential nuclei, (Rapp and Thomas, 2006). In addition to nucleation centers, the presence of water vapor and the low temperatures at the mid and high latitude mesopause during the summer months create conditions favorable for ice particle formation, (Avaste, 1993). Cold temperatures and water ice are known to be at the origin of another phenomenon called Noctilucent Clouds (NLC), (Schäfer et al., 2020), that are due to light scattered at the ice particles observed from the ground. More general, and when observed from space, the clouds of ice particles are denoted as polar mesospheric clouds, (Fritts et al., 2019).

The PMSE are formed through a process that involves the electrical charging of the ice particles and is for instance discussed by Rapp and Lübken (2004) and Latteck et al. (2021). They are strong radar echoes, and they result from reflections at inhomogeneities in the electron density when their spatial scales are of sizes comparable to half the radar wavelength. Constructive interferences of the reflections result in high backscattered power and narrowly peaked power spectra. Such strong echoes are typically from turbulence in the partially ionized upper atmosphere. The PMSE are in addition influenced by the presence of charged ice particles. The ice particles are spatially structured by the turbulence and as the ice particles collect ambient electrons when they are charged, they cause electron gradients to last longer and to form on smaller scales. The neutral atmospheric motion and dissipation of gravity waves at these altitudes are causes for the turbulence. The radar echoes in PMSE are stronger compared to normal incoherent scattering.

The EISCAT VHF radar used in our study is designed to measure the incoherent scatter, which comes from the small scale fluctuations of electrons in the ionospheric plasma. As the ionospheric electrons are exposed to the electromagnetic wave transmitted by the radar, the Thomson scattering scatters a small fraction back. The back scattered power is proportional to the electron density and the electron oscillations, which in turn are influenced by ion interactions. As a result, the spectra measured from incoherent scatter allow one to derive from the observed signal the electron density, and electron and ion temperatures, (Beynon and Williams, 1978). In their study, Rapp and Lübken (2004) elucidate the difference to PMSE, where PMSE are typically stronger than incoherent scatter located at the same altitude, and their spectra are more narrow. Observations with radars that also detect incoherent scatter offer the opportunity to measure the electron density in the vicinity of the PMSE.

Multi-layered Polar Mesospheric Summer Echoes have been the focus of several investigations, (Hoffmann et al., 2005), (Li et al., 2016), (Shucan et al., 2019). To simplify the exploration of PMSE multilayers, Jozwicki et al. (2021) conducted a study demonstrating the feasibility of distinguishing between images containing PMSE and those that do not, employing Linear Discriminant Analysis (LDA). Subsequently, in Jozwicki et al. (2022), a model built on random forests was employed to segment the PMSE signal from the incoherent scatter signal based on the power return in altitude. This model is utilized in the current paper for the pre-selection of data. An example of a PMSE occurrence with three distinct layers is depicted in Figure 1, inside of the red frame. Given the significance of electron density in PMSE formation, it is reasonable to expect a potential

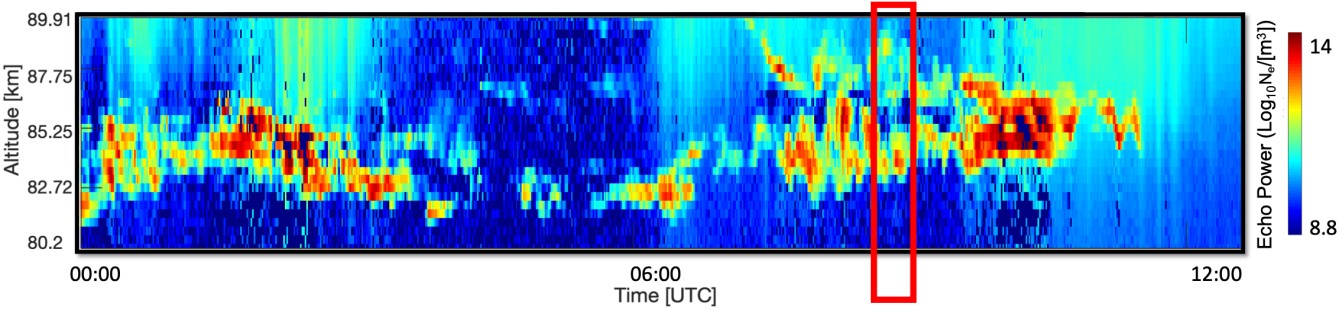

**Figure 1.** Data from EISCAT VHF from 16 July 2015 from 00:00 to 12:00, showing an example of a PMSE event that contains 3 multi-layers in the red frame.

influence of the solar cycle in it. Limited research has been conducted to examine the connection between multi-layered PMSE and the solar cycle.

We investigate PMSE observations with EISCAT VHF during the recent years. Our objective is to analyze the number of PMSE layers, their thickness, altitude, and general behavior during solar maximum and minimum, and to determine possible correlations between these variables and the electron density at ionospheric heights above PMSE. The study is organized as follows: In Sect. 2, we describe the methods and theories related to the pre-selection of the PMSE data, as well as the correlation coefficients employed to assess the significance of obtained results. In Section 3, we present and discuss the results. Finally, in Sect. 4, we summarize the conclusions drawn from this study.

## 2 Methods and Theory

In this section, we will describe our methodology for data selection, including the tools utilized. Furthermore, we will present the criteria used for identifying the different PMSE layers and the metrics employed for analyzing the collected data. In this study, we use recorded data from the EISCAT VHF radar located in Tromsø that operates at 224 MHz. The geographical coordinates of the EISCAT VHF radar are 69°35′N and 19°14′E; its geomagnetic latitude and longitude are respectively 66.73° and 102.18°.

### 2.1 Data selection

The Grand Unified Incoherent Scatter Design and Analysis Package (GUISDAP) is a software package used for processing and analyzing data from the EISCAT VHF incoherent scatter radar, (Lehtinen and Huuskonen, 1996). GUISDAP analysis fits the observed frequency spectrum received from each height with an incoherent scatter profile. The analysis returns the electron density based on the backscattered power, independently from the scattering process. The electron density parameter given by the analysis is proportional to the received echo power and therefore the strength of the PMSE.

**Table 1.** Data-set used for this study. The upper part of the table displays the dates and times selected for the solar maximum, and the lower part of the table is dedicated to the solar minimum. For each date, the corresponding sunspot number and the F10.7 cm flux is displayed. The F10.7 cm solar flux is given in $W.m^{-2}.Hz^{-1}$. The date and time format are given respectively by the dd/mm/yyyy and the ...h...m format.

| | F10.7 cm Flux | Sunspot Number | Year | Date | Start time | End time | Observation Hours per Day | Observation Hours per Year | Observation Hours per Solar Max. or Min. | Total of Observation Hours |
|---|---|---|---|---|---|---|---|---|---|---|
| **Solar Maximum** | 9.95000e-21 | 90.9 | 2013 | 27/06/2013 | 07h02m | 10h58m | 03h56m | 57h52m | 130h18m | 230h32m |
| | 1.01000e-20 | 90.9 | | 28/06/2013 | 07h02m | 12h58m | 05h56m | | | |
| | 1.19900e-20 | 94.6 | | 09/07/2013 | 00h00m | 00h00m | 24h00m | | | |
| | 1.17900e-20 | 94.6 | | 10/07/2013 | 00h00m | 00h00m | 24h00m | | | |
| | 9.91000e-21 | 112.6 | 2014 | 23/07/2014 | 00h00m | 09h26m | 09h26m | 09h26m | | |
| | 1.01000e-20 | 68.3 | 2015 | 15/07/2015 | 08h00m | 00h00m | 16h00m | 63h00m | | |
| | 9.96000e-21 | 68.3 | | 16/07/2015 | 00h00m | 00h00m | 24h00m | | | |
| | 9.74000e-21 | 68.3 | | 17/07/2015 | 00h00m | 23h00m | 23h00m | | | |
| **Solar Minimum** | 6.70000e-21 | 3.7 | 2019 | 18/06/2019 | 06h59m | 00h00m | 17h00m | 59h13m | 100h14m | |
| | 6.80000e-21 | 3.7 | | 19/06/2019 | 00h00m | 12h59m | 12h59m | | | |
| | 6.80000e-21 | 3.5 | | 04/07/2019 | 07h07m | 12h21m | 05h14m | | | |
| | 6.70000e-21 | 3.4 | | 20/08/2019 | 00h00m | 00h00m | 24h00m | | | |
| | 6.90000e-21 | 9.0 | 2020 | 06/07/2020 | 07h58m | 09h08m | 01h06m | 41h01m | | |
| | 6.80000e-21 | 9.0 | | 07/07/2020 | 00h00m | 11h59m | 11h59m | | | |
| | 6.70000e-21 | 9.0 | | 08/07/2020 | 00h00m | 11h59m | 11h59m | | | |
| | 6.90000e-21 | 9.0 | | 09/07/2020 | 00h00m | 11h58m | 11h58m | | | |
| | 6.90000e-21 | 9.0 | | 10/07/2020 | 08h00m | 11h59m | 03h59m | | | |

We downloaded over 230 hours of recorded data via the Madrigal website. This corresponds to 17930 data points, with the details provided in Table 1. The EISCAT VHF radar utilizes many different experimental modes to collect data. The utilized pulse coding for the PMSE measurements we analyzed is referred to as 'Manda'. Some parameters of the EISCAT VHF radar using the 'Manda' experiment are listed in Table 2. Detailed information regarding this experiment can be found on the EISCAT website (https://eiscat.se/scientist/document/experiments/). For this study, we specifically analyzed data obtained using the 'Manda' experiment, because it is designed to detect low-altitude signals and layers in the mesosphere. We chose a time resolution of 60 seconds and a height resolution of 0.360 km.

We employed EISCAT VHF frequencies over UHF frequencies due to the latter exhibiting a lower recorded amount of PMSE compared to VHF frequencies. As the Heating experiment is known to influence the back-scattered power (also known as echo power) of the PMSE, (Belova et al., 2003), we carefully selected data from the days when the Heating experiment was not performed. This enabled us to compare electron densities at 92 km altitude alongside echo power at PMSE altitudes.

The data was carefully selected to encompass the solar maximum and solar minimum phases of the solar cycle. For the purpose of this study, we do not require an absolute value of PMSE strength, thus, we do not perform all the steps that would be necessary to obtain the absolute radar reflectivity as per the study by Hocking et al. (1986).

**Table 2.** Some parameters of the EISCAT VHF radar, the source of data for this paper. More information about the EISCAT documentation and radar system parameters can be found at: https://eiscat.se/scientist/document/experiments/.

| EISCAT VHF parameters | |
|---|---|
| Frequency | 223.4 MHz |
| Wavelength | 1.34 m |
| Bragg scale | 0.67 m |
| Peak power | 1.2 MW |
| Transmitted pulse scheme | Manda v 4.0 |
| Interpulse period | 1.5 ms |
| Time resolution | 4.8 s |
| Range resolution | 360 m |
| Spectral resolution | 2.6 Hz |
| Antenna Elevation | 90 deg, zenith |

To investigate the behaviour of the ionosphere in relation to PMSE, we compared the echo power for PMSE altitudes between 80 and 90 km, with the electron density at 92 km ionospheric altitude. We used the electron density at 92 km altitude as a reference as it was the closest to the PMSE altitudes and the results were similar for altitudes of 92, 95, and 100 km.

## 2.2 Data processing

In this paper, we consider two variables: echo power and electron density. Both are measured in base 10 logarithmic units of the number of electrons per cubic meter. The number of electrons per cubic meter is proportional to the back-scattered power for incoherent scatter, where the back-scattered power is defined as the amount of power in the scattered signal received by the antenna. We define the back-scattered power at 92 km altitude as electron density. The back-scattered power at PMSE altitudes, between 80 and 90 km altitude, is defined as echo power.

We selected the PMSE data between 80 and 90 km altitude by using a segmentation model from the study by Jozwicki et al. (2022). The segmentation model used random forests on a set of hand-crafted features to segment the PMSE data from the background. Random forests is a machine learning algorithm used for both classification and regression. In this algorithm, a number of decision trees are used during training phase to make predictions. On the output from the segmentation model, we applied a threshold to ensure that only PMSE data were retained for further analysis. This thresholding technique was also employed in the study by Shucan et al. (2019), where they used an echo power threshold $N_e > 2.6 \times 10^{11} m^{-3}$, and in the study by Rauf et al. (2018b), where the authors used a threshold $N_e > 5.0 \times 10^{10} m^{-3}$. We were able to use a lower threshold of $N_e > 3.2 \times 10^{10} m^{-3}$ (which is equivalent to 10.5 in base 10 logarithmic units of the number of electrons per cubic meter) as the segmentation model from the study by Jozwicki et al. (2022) had successfully removed almost all non-PMSE data. This

enabled us to retain a large amount of PMSE data per number of hours of observation, in comparison to the findings of Shucan et al. (2019) and Rauf et al. (2018b).

## 2.3 Detection of PMSE multi-layers

After processing the data at PMSE altitudes as described in Sect. 2.2, we aimed to detect the start and end of each PMSE layer in altitude. To achieve this, we utilized a method used in the study by Hoffmann et al. (2005) and Shucan et al. (2019). This method involves defining the start of a layer each time the threshold for echo power is exceeded, and the end of the layer when the echo power falls below the given threshold. The time intervals and the corresponding altitude intervals associated with the start and end of each layer were recorded. During solar maximum conditions, we observed a maximum of six layers. In this study, we decided to ignore multi layers with more than 4 layers, as their occurrence rates were low. For instance, we observed 13 occurrences of 5 multi layers in the whole data-set, and 2 occurrences of 6 multi layers. In Table 3, we show the occurrences of monolayer and multilayer PMSE events, observed during the solar minimum and solar maximum phases, with each occurrence corresponding to a 1-minute interval.

## 2.4 Data analysis

In this study, we perform comparisons between the different mono and multi layers of PMSE by using a number of parameters. The parameters included the starting and ending altitude intervals of the layer, the layer thickness (calculated as the difference between the start and end altitude interval), the mean altitude interval that corresponds to the middle of the layer, the echo power in the mean altitude interval inside the PMSE, the altitude of the mean altitude interval, the layer's time interval, the UTC time associated with the time interval, the number of layers present in the time interval, and the electron density at 92 km altitude.

In order to investigate different PMSE properties, we use the Pearson correlation coefficient and the Spearman's rank correlation coefficient to calculate the correlations between the different parameters, (Wilks, 1995), (Myers and Well, 2003). The Pearson correlation coefficient is used to measure how strong and in what direction two variables are related in a linear way, (Wilks, 1995). For two random variables X and Y, the Pearson correlation coefficient is defined as follows, (Wilks, 1995):

$$r_{Pearson}(X,Y) = \frac{cov(X,Y)}{\sigma_X \sigma_Y} \tag{1}$$

Where $\sigma_X$ and $\sigma_Y$ are the respective standard deviations of X and Y, and $cov$ is the covariance.

The Spearman's rank correlation coefficient is a measure of the strength and direction of the relationship between two variables. It is similar to the Pearson correlation, but instead of measuring the linear relationship between two variables, it measures the monotonic relationship between them. The Spearman's rank correlation coefficient is obtained by calculating the Pearson correlation between the ranked values of the variables, (Myers and Well, 2003). To compute the Spearman correlation coefficient, for a sample size n, the raw scores Xi and Yi are converted into their rank values $rg_X$ and $rg_Y$. After that, the Spearman correlation coefficient is then computed as follows:

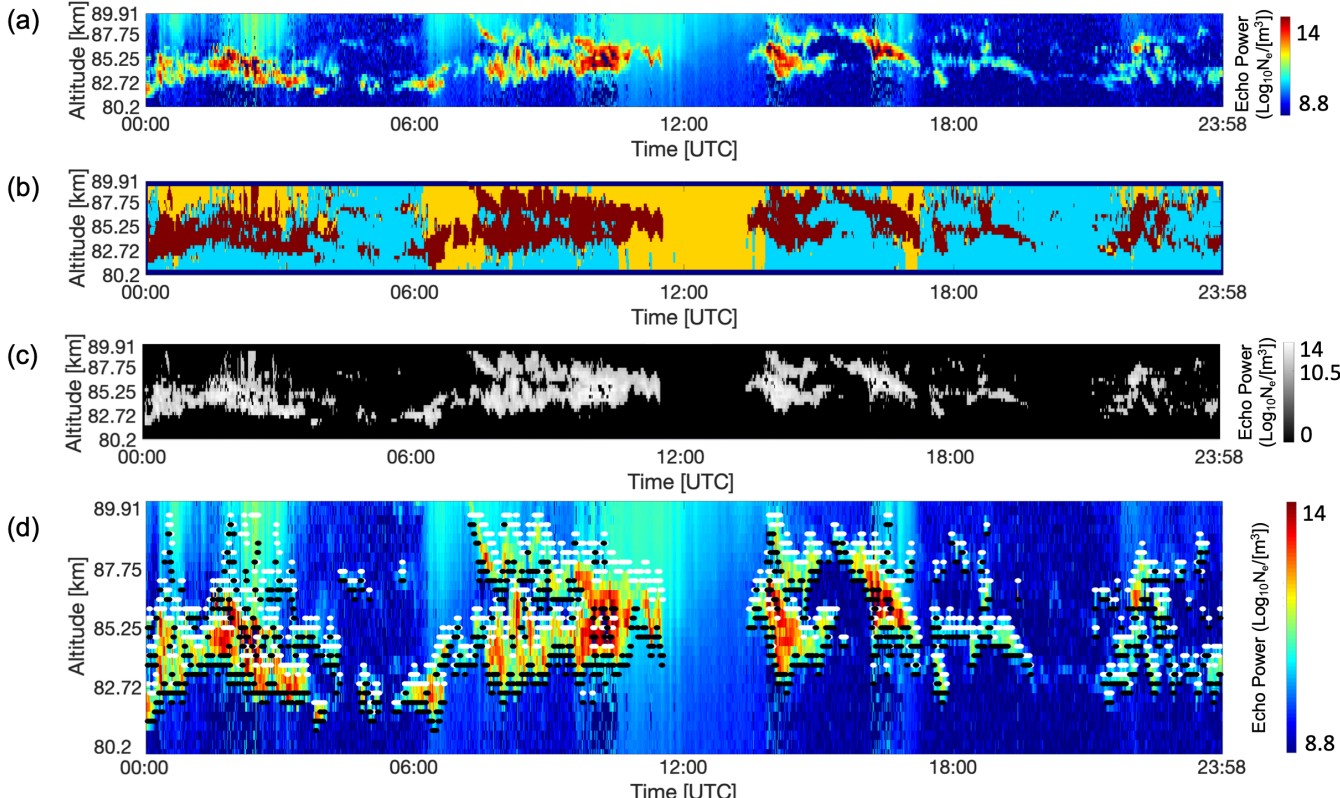

**Figure 2.** Figure illustrating the process of the layer detection. **(a)** shows the original data for the 16 July 2015 between 00:00 and 23:58. **(b)** shows the output from the classification model used from Jozwicki et al. (2022). Dark red represents areas labeled as PMSE, cyan represents areas of the data labeled as background noise and yellow represents areas labeled as ionospheric background. **(c)** represents the data labeled as PMSE in dark red from sub Fig. 2(b), onto which we applied the threshold described in Sect. 2.2 to make sure we have only PMSE data left. Finally, **(d)** represents the detected beginning and end of layers respectively represented with white and black points, overlayed on the original data.

$$r_{Spearman} = \frac{cov(rg_X, rg_Y)}{\sigma_{rg_X} \sigma_{rg_Y}} \tag{2}$$

Where $\sigma_{rg_X}$ and $\sigma_{rg_Y}$ are the standard deviations of the rank variables, and cov($rg_X, rg_Y$) is the covariance of those rank variables.

In this analysis, we calculated the statistical significance of our results using the p-value (t-test), which are listed in Tables B2, B4 and B3 in the Appendix. P-values are used to determine whether the obtained results are different enough to be judged statistically significant or not, using the means, variances, and populations of the given variables. If the p-value falls below the significance level (alpha), the given result is considered to be statistically significant. Testing the statistical significance of results comes with various confidence levels (90%, 95%, and 99%), which depend on the chosen significance level (with

**Table 3.** This table displays the number of occurrences and approximate percentage of occurrence for each of the mono and multi layers in our data-set. The data is separated by solar maximum and solar minimum. For both solar maximum and solar minimum, the approximate percentage of occurrence for 5 multi layers or more is below one percent. Therefore, the analysis in this study is limited to PMSEs with up to four multi-layers.

| | | Number of Occurrences | Total Number of Occurences per Sol Max. or Min. | Approximate Pencentage of Occurence |
|---|---|---|---|---|
| Solar Maximum | Mono Layers | 3077 | | 51 |
| | 2 Multi Layers | 2233 | | 37 |
| | 3 Multi Layers | 597 | | 10 |
| | 4 Multi Layers | 81 | 5996 | 1 |
| | 5 Multi Layers | 6 | | <1 |
| | 6 Multi Layers | 2 | | <1 |
| | 7 Multi Layers | 0 | | 0 |
| Solar Minimum | Mono Layers | 1399 | | 51 |
| | 2 Multi Layers | 935 | | 34 |
| | 3 Multi Layers | 328 | | 12 |
| | 4 Multi Layers | 67 | 2736 | 2 |
| | 5 Multi Layers | 7 | | <1 |
| | 6 Multi Layers | 0 | | 0 |
| | 7 Multi Layers | 0 | | 0 |

corresponding significance levels of 0.1, 0.05, and 0.01). It is commonly accepted that a p-value below $\alpha = 0.05$ is indicative of statistical significance. However in this study, we are analyzing a multi parameter dataset, which is why we chose a lower threshold of $\alpha = 0.0001$, which is two orders of magnitude more selective.

## 3 Results and Discussion

In this section, we will discuss our results which are organized into multiple parts. Firstly, we will discuss the distributions of
155 a few variables, which will be presented using histograms. Subsequently, we will analyze the correlation coefficients that we have computed for the different variables.

## 3.1 Height distribution of PMSE layers

Our study focuses on observations from the summer mesopause during solar maximum in years 2013 to 2015, and solar minimum in years 2020 and 2021. The average peak altitude of PMSE height distribution, considering all PMSE detections, is higher during solar maximum than during solar minimum (see Fig. 3). The averaged mean altitude values of all the separate layers in the different sets of 2 multilayers, 3 multilayers and 4 multilayers, are shown in the Appendix section, in Fig.A1 and Fig.A2.

When considering the mean altitude values of individual layers within the sets of 2, 3, and 4 multilayers, a trend is seen in Fig. 4 and Fig. 5. In these figures, the color scheme has the red distribution representing the highest-altitude layer (the topmost layer), followed by the green distribution for the second highest layer, the blue distribution for the third highest layer, and the magenta distribution for the fourth highest layer. Additionally, when two layers' altitude distributions overlap, an intermediate color arises to represent this overlap. The p-values for all possible combinations of these individual layers, as shown in Fig. 4 and Fig. 5, can be found in Table B1 in the Appendix. Upon decomposing the multilayer sets into individual layers, one can see that in both solar maximum and solar minimum conditions, the altitude of the top layer increases as the number of multilayers increases. This pattern holds true for the second and third highest layers as well.

Our study confirms the findings of Hoffmann et al. (2005) regarding the altitude of the observed mono and multiple layers. Hoffmann et al. (2005) examined the occurrence and mean altitude of PMSE layers, and performed microphysical model simulations. They proposed that the observed multiple PMSE layer structures are mainly caused by the layering of ice particles due to subsequent nucleation cycles. They reported that mono layers occurred at an average altitude of 84.8 km, and our results show that the mean altitude of mono layers was 85.21 km for solar maximum and 84.46 km for solar minimum. Our mean altitude of 84.83 km is consistent with the results of Hoffmann et al. (2005). Furthermore, they observed that in a set of two multilayers, the lower layer occurs at a mean altitude of 83.4 km, and the upper layers occurs at a mean height of about 86.3 km, which is consistent with our findings. In fact, we found that in a set of two multilayers, the lower layer happens at a mean altitude of 83.74 km for solar maximum and 82.90 km for solar minimum, which results in an average of 83.32 km. Additionally, the upper layer occurs at an average altitude of 86.71 km for solar maximum and 85.97 km for solar minimum, which results in an average altitude of 86.34 km over the whole solar cycle. For this reason, we can note a similar observation as Hoffmann et al. (2005) did in their study, that the altitude of the lower layer is in good agreement with the mean altitude of NLC from lidar observations made by Fiedler et al. (2003) at ALOMAR (at 69°16'42.0"N 16°00'29.0"E, i.e. close to EISCAT), where the mean altitude of NLC was found to be of about 83.3km. When examining the lowest layer in various multilayer sets in Fig. 4 and Fig. 5 (not limited to a set of just two multilayers, as discussed earlier), one can notice that the lowest layer almost always aligns with the NLC altitude as reported by Fiedler et al. (2003). Finally, Hoffmann et al. (2005) observed that mono layers occurred 50.1%, double layers 36.6%, and multi layers with more than 2 layers 13.3%, during both solar maximum and minimum periods. Our study indicates that mono layers were observed at a rate of 51% in both solar maximum and minimum, while double layers occurred at a rate of 37% in solar maximum and 34% in solar minimum. Furthermore, we found that the

occurrence rate for multi layers with 3 and 4 layers combined was more than 11% in solar maximum and more than 14% in solar minimum.

The solar maximum phase is characterized by an increased number of sunspots and higher levels of ultraviolet radiation compared to the solar minimum phase. The F10.7 flux is often used as a proxy for the level of solar activity and, more specifically, the amount of ultraviolet radiation. The K index describes geomagnetic activity and potentially corresponds to

particle precipitation. Shucan et al. (2019) found that PMSE mono, double, and triple layer occurrence ratios are positively correlated with the K index. Also, Shucan et al. (2019) mentions that the PMSE triple layer occurrence ratio shows a negative correlation with F10.7. Zhao et al. (2020) reported a positive correlation between the temperature of the mesopause and the F10.7 flux. They found that the temperature of the mesopause is decreasing with time over 18 year long investigation (from 0 to -0.14 K/year), which could affect the formation of PMSE. They also found that the height of the mesopause is decreasing

with time at polar latitudes, which could potentially impact the height of PMSE.

Lübken et al. (2021) show in their study that over time the ice particles are increasing in size. In Fig. 3, we can see that the altitude of the PMSE layers is on average lower for solar minimum compared to solar maximum. This could be due to the fact that the ice particle sizes increase with time over years, and our selected date for the solar maximum are anterior to the selected dates for the solar minimum. Considering these findings, the small difference in the altitude of the layers that we noted may

be due to trends not related to solar cycle effects. Therefore, it appears that factors other than the sole influence from the solar cycle play a significant role in the altitude of PMSE. Finally, further investigations, comparing the next solar maximum to the previous one, might bring more clarity in the understanding of the influence from the solar cycle alone.

### 3.2 Distribution of the electron density

In the next step, we investigate how the distribution of the PMSE layers changes with ionization. We consider the electron

densities observed above the PMSE and ignore specific causes of ionization in this study. All the observed electron densities are summarized in Fig. 6; they range from 8.9 to 11.7 electrons per cubic meter in base 10 logarithmic unit during solar maximum and their mean value is slightly higher during solar maximum. Specifically, multi-layer PMSEs with 2 layers exhibit the highest average corresponding electron density, reaching 10.47 electrons per cubic meter in base 10 logarithmic unit as one can see from Fig. 7. In contrast, the mono layers during solar minimum have the lowest average corresponding electron density,

with a value of 10.15 electrons per cubic meter in base 10 logarithmic unit, as displayed in Fig. 8. It is worth noting that, for both solar maximum and solar minimum periods, the mono layers corresponded to the lowest average electron density of their respective seasons. However, it is important to bear in mind that this trend is weak and that some P-values corresponding to the different combinations of layers in Fig. 7 and Fig. 8 are greater than 0.05, as shown in Table B2. A plausible argument could be made that higher electron densities at ionospheric altitudes might be necessary to observe multi-layered PMSEs.

During solar maximum, we observe a wider range of electron densities compared to solar minimum when PMSE are present, particularly at higher electron densities. This variation in electron densities may explain why the mean electron density at an altitude of 92 km is higher during solar maximum than solar minimum during PMSE events. Additionally, our analysis reveals that the standard deviation of electron densities decreases with increasing number of layers, with mono layers exhibiting

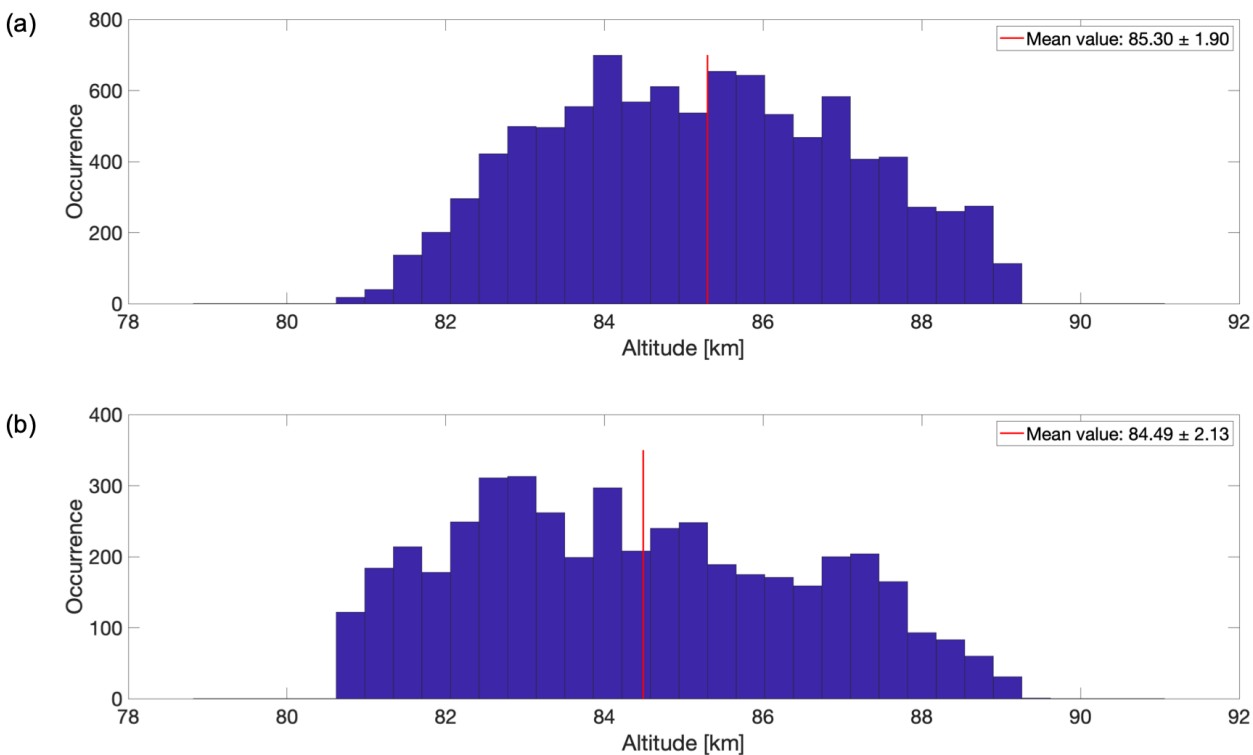

**Figure 3.** Altitude distribution of the data for the **(a)** solar maximum and **(b)** solar minimum. Each subplot was its respective mean altitude represented with a red line on the graph, and specified in the legend together with one standard deviation.

the largest standard deviations and 4-layer systems exhibiting the smallest standard deviations, for both solar maximum and
minimum conditions.

### 3.3 Distribution of the echo power

As discussed in Sect. 2.2 we classified the data using the classification model of Jozwicki et al. (2022) and applied a threshold to identify PMSE. Specifically, we considered all echo power values above a threshold of 10.5 electrons per cubic meter in base 10 logarithmic unit as PMSE. This explains the absence of values below 10.5 on the horizontal axis of Fig. 9, Fig. 10 and
Fig. 11. Figures have been generated to visualize individual layers within the various sets of 2, 3, and 4 multi-layers seen in Fig. 10 and Fig. 11. This approach mirrors the technique employed in Fig. 4 and Fig. 5. However, since the separation of layers did not yield additional information, we have chosen to retain the averaged representations of all multi-layers combined, as depicted in Fig. 10 and Fig. 11.

In Fig. 9, it is evident that the average echo power in PMSE is higher during solar maximum than solar minimum. We
noticed a greater distribution of higher values of echo power during solar maximum as compared to solar minimum, which leads to higher mean value during the solar maximum. Further, in Fig. 10, we observe that the average echo power decreases

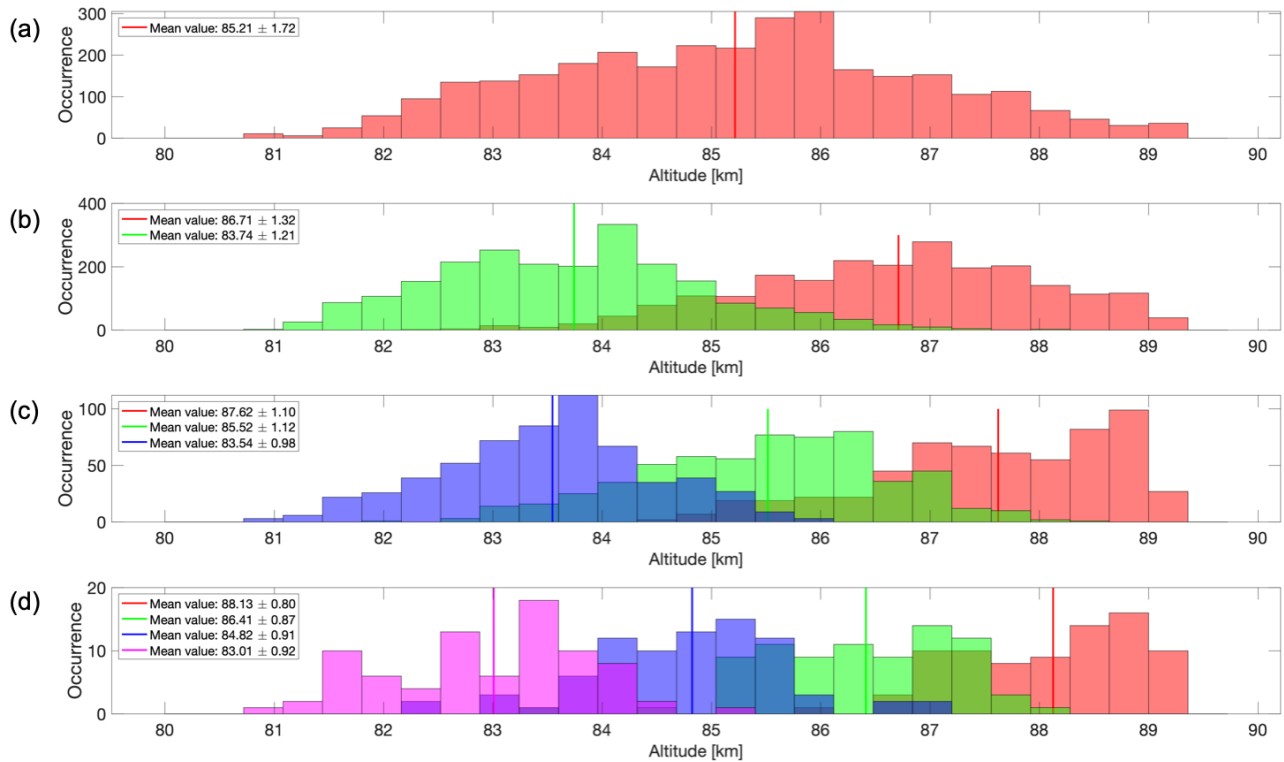

**Figure 4.** Altitude distribution of the data during solar maximum for **(a)** mono layers, **(b)** multi layers with 2 layers, **(c)** multi layers with 3 layers, and **(d)** multi layers with 4 layers. In each figure, the color scheme of the distributions indicates altitude order: red for the highest layer, green for the second highest, blue for the third highest, and magenta for the fourth highest. Intermediate colors represent overlapping altitude distributions. The legend displays the mean value and one standard deviation for each distribution.

as the number of multi-layers increase for solar maximum and the individual layers considered. This indicates that a single mono-layer has a higher echo power than the individual layers of two multi-layers, which in turn have a higher echo power than the individual layers of three multi-layers, and so on. However, during solar minimum as shown in Fig. 11, this trend is less evident, and we do not see a clear decrease in echo power with increasing number of layers.

### 3.4 Distribution of the thickness

In our study, we determined the thickness of the PMSE layers based on the number of neighboring data points, or altitude channels exceeding the echo power threshold described in Sect. 2.2. Each data point or altitude channel corresponds to a distance of 360m. As shown in Fig. 12, the average thickness of the layers is higher during solar maximum, with an average of 1.59 km, compared to solar minimum, where the average thickness is 1.32 km. When we examine the mono and multi-layer cases in more detail, as shown in Fig. 13 and Fig. 14, we observe that the average thickness decreases as the number of layers increases. This means that a mono-layer will be thicker than a layer belonging to a set of 2 multi-layers, which in turn will be

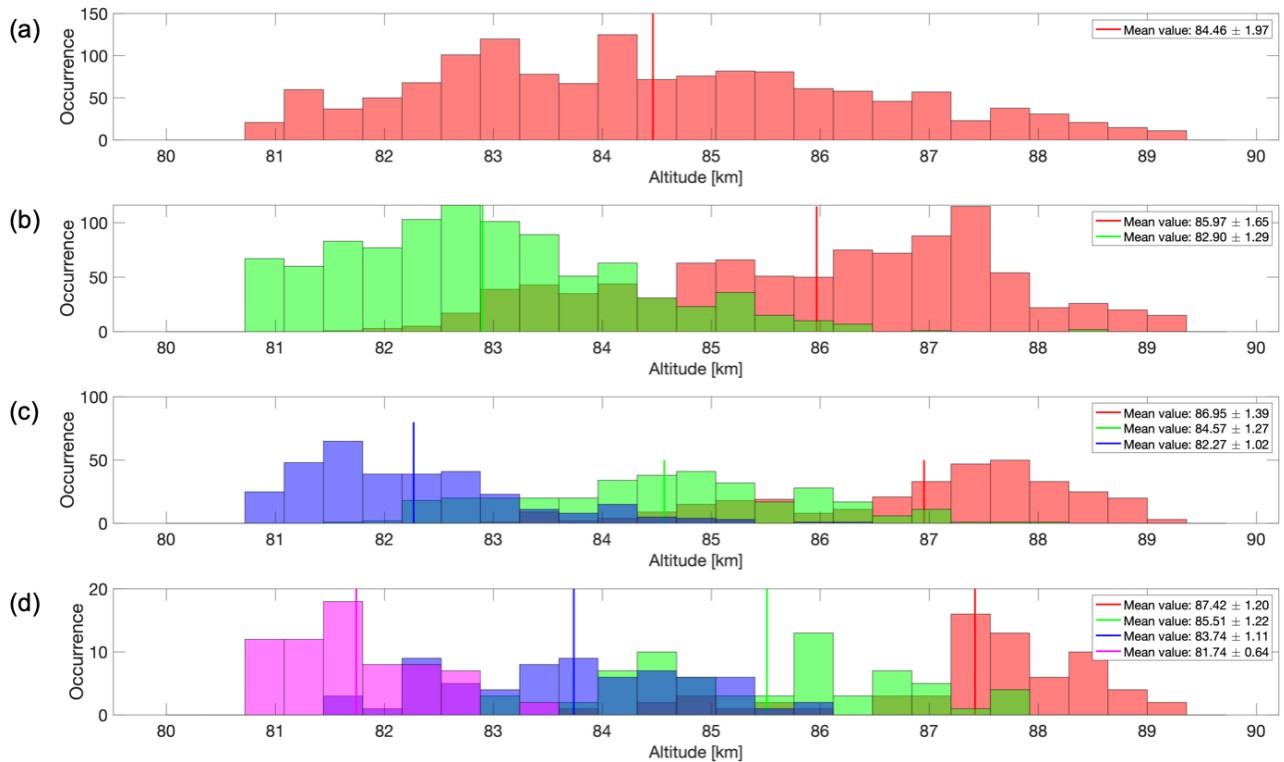

**Figure 5.** Altitude distribution of the data during solar minimum for **(a)** mono layers, **(b)** multi layers with 2 layers, **(c)** multi layers with 3 layers, and **(d)** multi layers with 4 layers. In each figure, the color scheme of the distributions indicates altitude order: red for the highest layer, green for the second highest, blue for the third highest, and magenta for the fourth highest. Intermediate colors represent overlapping altitude distributions. The legend displays the mean value and one standard deviation for each distribution.

thicker than a layer in a 3 multi-layer case, and so on. Figures have been generated to visualize individual layers within the various sets of 2, 3, and 4 multi-layers seen in Fig. 13 and Fig. 14. This approach mirrors the technique employed in Fig. 4 and Fig. 5. However, since the separation of layers did not yield additional information, we have chosen to retain the averaged representations of all multi-layers combined, as depicted in Fig. 13 and Fig. 14. The highest average layer thickness is obtained during solar maximum for mono-layers with an average of 2.15 km, while the lowest average of 0.87 km is obtained during solar minimum, for 4 multi-layers.

A comparison can be drawn between the thickness of NLCs and PMSEs. Although the formation mechanisms of these two phenomena differ, there is a shared population of ice particles that contribute to both. Therefore, it is worthwhile to explore the potential similarities and differences between them. Lübken et al. (2009) found that NLCs have higher brightness at lower altitudes, while Schäfer et al. (2020) analyzed 182 hours of LIDAR data and found that NLCs occur more than half of the time (57.2%), in thick layers of more than 1 km. In our study, we analyzed 7790 instances of PMSEs with 3 or more altitude channels. Knowing that one altitude channel corresponds to 360m, 3 altitude channels or more indicate a PMSE thickness

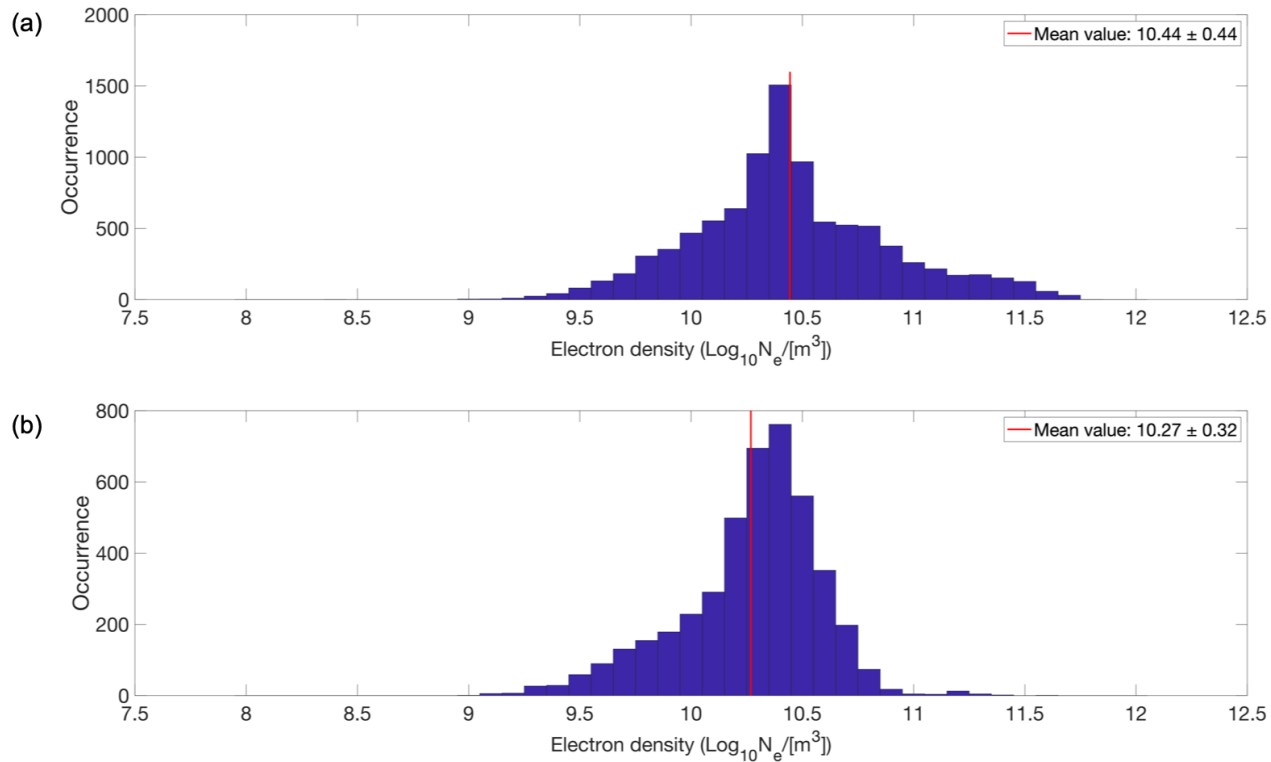

**Figure 6.** Electron densities at 92 km altitude for all layers during **(a)** solar maximum and **(b)** solar minimum. Each subplot was its respective mean electron density represented with a red line on the graph, and specified in the legend together with one standard deviation.

of at least 1.08 km. Our findings show that 54.64 percent of PMSE occurrences resulted in thick layers of 1.08 km or more. These results are consistent with those of Schäfer et al. (2020), where they reported that 57.2 percent of NLC occurrences were observed in thick layers of 1 km or more. Additionally, Schäfer et al. (2020) classified the NLCs they observed into 10 subcategories and found that the most frequently occurring subcategory consists of thick layers composed of multiple multi-layers, with an occurrence rate of 20.5%. They report that each of the multi layers move in parallel with each other. This

implies that there is a similar movement in the vertical displacement of the multi layers. If we consider all types of multi layers, mentioned by Schäfer et al. (2020), this percentage increases up to 27.6%. In our study, multi layers happen half of the time, with an approximate occurrence rate of 49%. Therefore our results differ from the ones of Schäfer et al. (2020) when it comes to the occurrence rate of multi layers, which may be explained by some of the differences in the formation and measurement of the two phenomena.

Gravity waves are thought to play a role in the formation of PMSE by generating neutral turbulence in the mesosphere. The complex dynamics and structuring because of shear instabilities and breaking of the gravity waves are derived, for example, from polar mesospheric cloud observations, and can generate turbulence at PMSE altitudes, (Fritts et al., 2019). This turbulence can lead to small-scale variations in the electron density, which can create the conditions necessary for PMSE to form, (Rapp

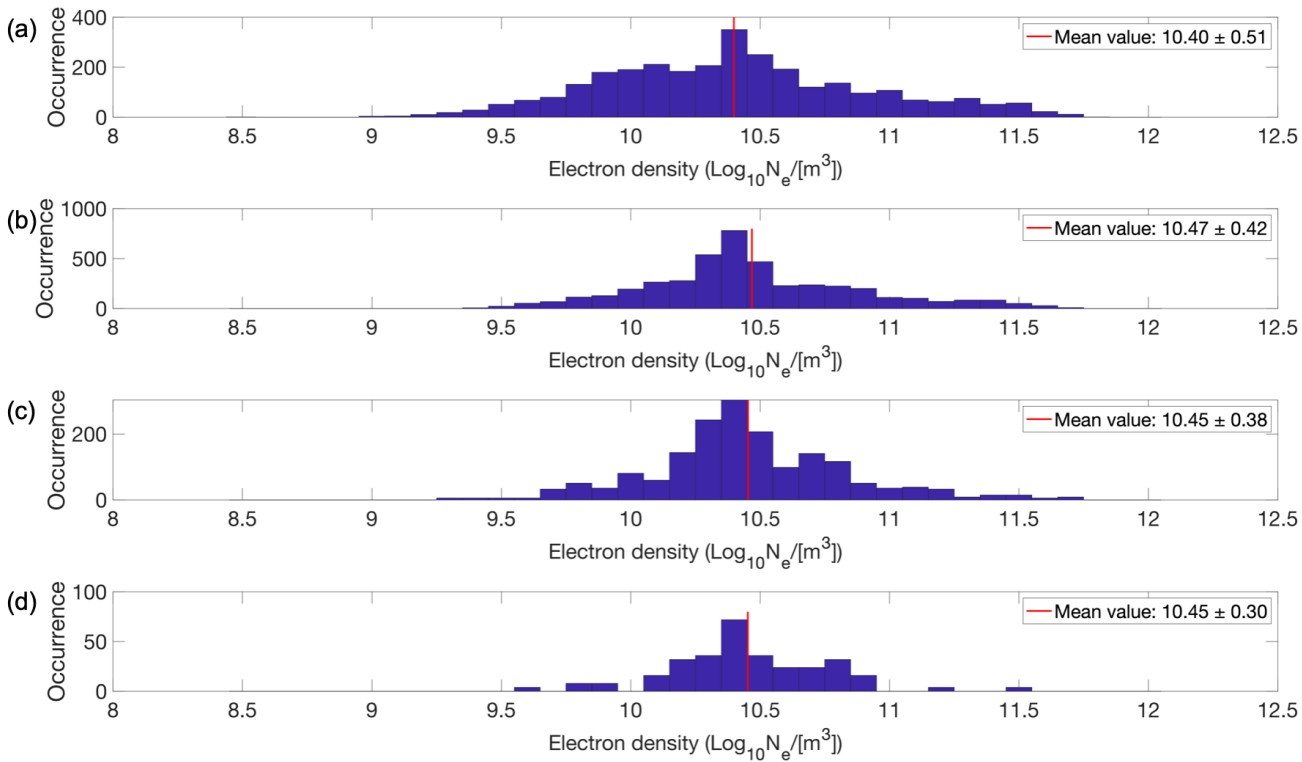

**Figure 7.** Electron density at 92 km altitude during solar maximum for **(a)** mono layers, **(b)** multi layers with 2 layers, **(c)** multi layers with 3 layers, and **(d)** multi layers with 4 layers. Each subplot was its respective mean electron density represented with a red line on the graph, and specified in the legend together with one standard deviation.

and Lübken, 2004). Therefore, understanding the characteristics of gravity waves and their effects on the neutral atmosphere
is essential for understanding the formation of PMSE.

Li et al. (2016) developed a two-dimensional theoretical model to explore the creation process of multi layered PMSE. The aim of the proposed model was to consider how gravity waves could cause movement of ice particles through collisions with the neutral atmosphere. Their model was able to simulate the presence of gravity waves by assigning both vertical and horizontal wavelengths. The ice particles are considered to be spherical, and their size does not vary during the simulations.
This means that processes such as growth, sedimentation or sublimation are not taken into account in their model. In their first experiment, Li et al. (2016) fixed the particle size at 10 nm and varied the vertical wavelength of gravity waves to 3 km, 4 km, and 5 km. Only one wavelength was considered at a time, when varying the vertical wavelength. They observed a decrease in the number of layers as the vertical wavelength increased. Also, the thickness of the layers increased as the number of layers decreased. Our results on thickness distribution as shown in Fig. 12, Fig. 13 and Fig. 14 show similar trends. We found that the
average thickness of mono layers was higher than that of multi layers, and the thickness decreased with an increasing number

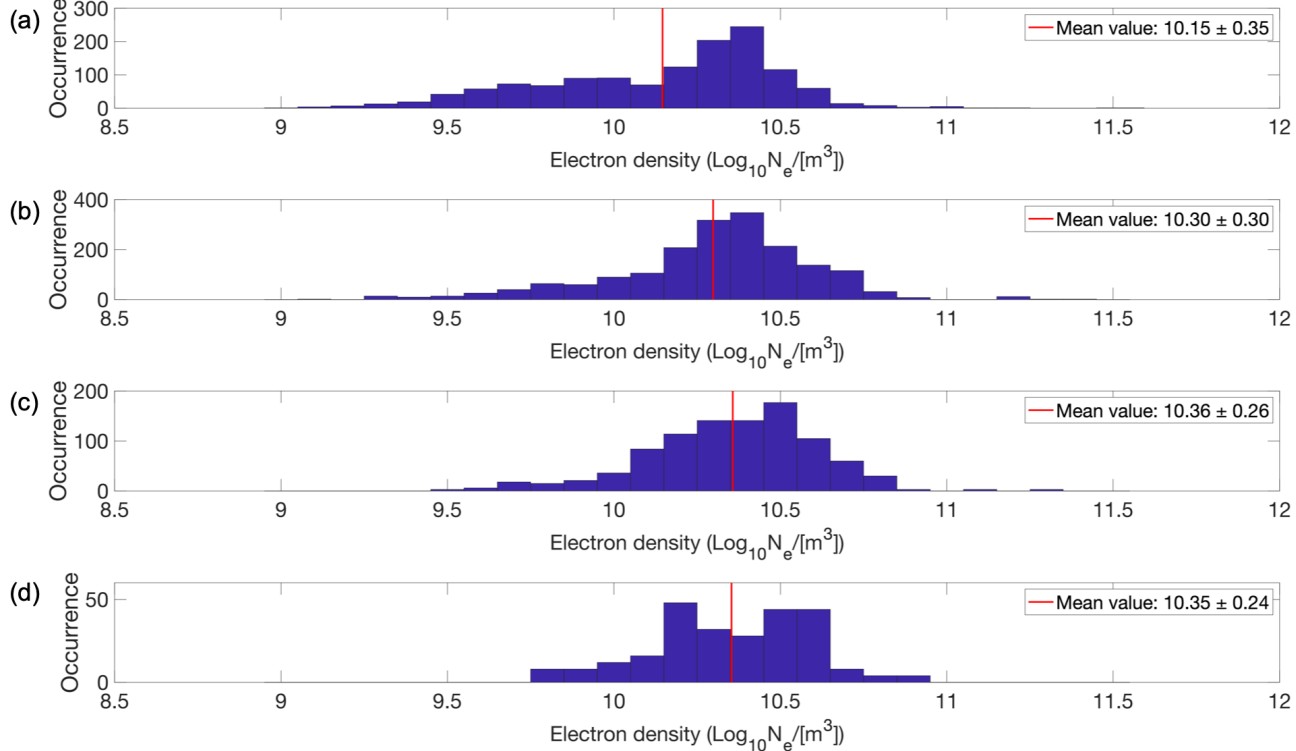

**Figure 8.** Electron density at 92 km altitude during solar minimum for **(a)** mono layers, **(b)** multi layers with 2 layers, **(c)** multi layers with 3 layers, and **(d)** multi layers with 4 layers. Each subplot was its respective mean electron density represented with a red line on the graph, and specified in the legend together with one standard deviation.

of multi layers. One possible hypothesis that can be drawn is that the thickness of the layers could be related to the vertical wavelength of gravity waves, with higher wavelengths producing thicker layers.

In another experiment in Li et al. (2016) study investigated the effect of varying ice particle size while fixing the vertical wavelength of gravity waves at 4km. They used particle sizes of 10 nm, 20 nm, and 30 nm and found that the altitude of the layers decreased more rapidly and their formation became more challenging with increasing particle size. Also, once the turbulence stopped, the larger ice particles took longer to go back to a neutral homogeneous state. It is worth noting that their model does not consider the growth, sedimentation, and sublimation processes, so these findings should be considered as preliminary hypotheses. Li et al. (2016) also reported the observation of preferred altitudes for each multi layer formation, which depended on the size of the ice particles. Potential mechanisms for ice formation at upper mesospheric altitudes that could be affected by the solar cycle are unknown to the authors, but this is something to investigate in a future study.

Neutral air turbulence which is a key factor in PMSE formation can be generated by wind shears. Singer et al. (2012) found that westward winds are increasing below an altitude of about 85 km, while eastward winds are increasing above 85 km, particularly during summer. They also found that at an altitude of about 75 km, the long-term trend of zonal winds corresponds

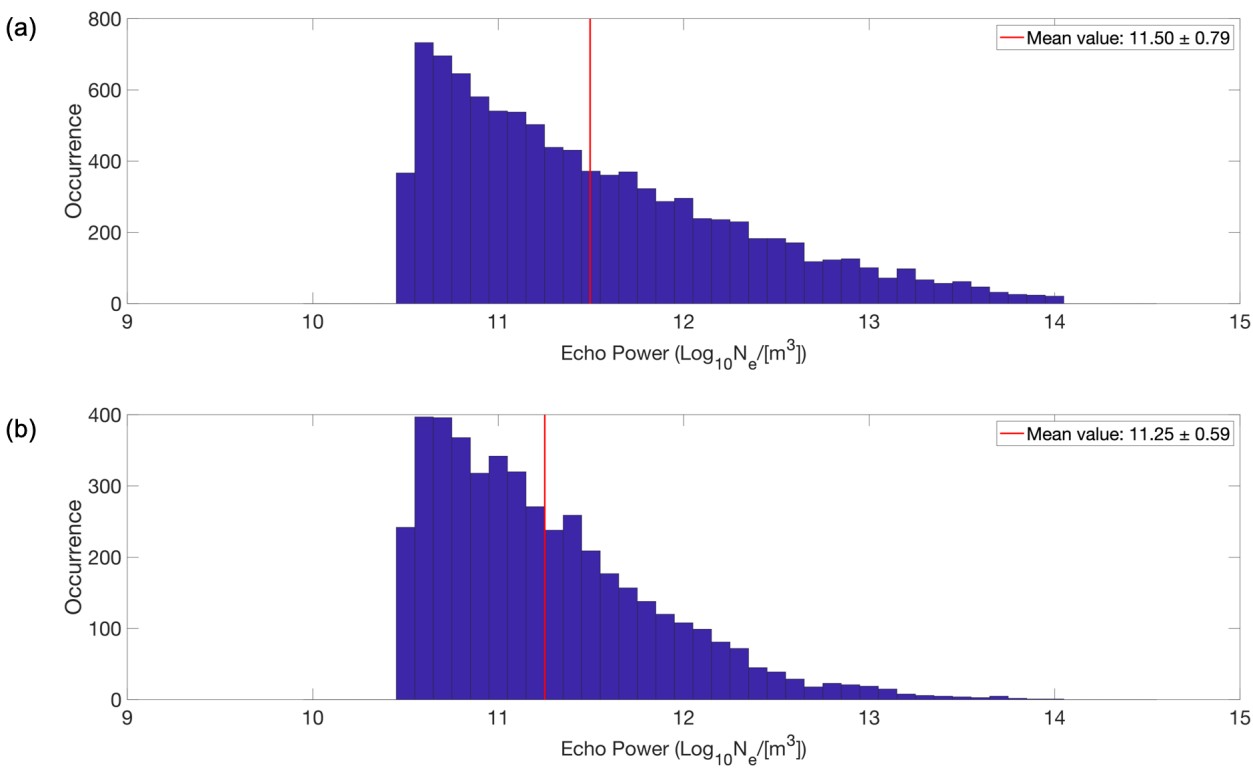

**Figure 9.** Echo power in the PMSE for all layers during **(a)** solar maximum and **(b)** solar minimum. Each subplot has its respective mean echo power represented with a red line on the graph, and specified in the legend together with one standard deviation.

to increased activity of gravity waves with periods of 3 to 6 hours at altitudes between 80 km and 88 km. Severe solar proton
events cause eastward winds to increase above an altitude of about 85 km. This behavior of winds and their effects at PMSE altitudes may be another key to a better understanding of the formation of multi-layered PMSE.

### 3.5 Correlations

In this section, we will analyze the correlation between several parameters, namely electron density, echo power, thickness, and altitude. Table 4 shows both correlation coefficients for all layers together, for the solar maximum on the lower portion of
the table, and for the solar minimum on the upper portion of the table. Table 5(a) shows the results of the Pearson correlation coefficient only, for mono and multi layers separately, and for solar maximum and minimum. Table 5(b) shows the results of the Spearman's rank correlation coefficient only, for mono and multi layers separately, and for solar maximum and minimum. For simplicity, in all the mentioned above tables, the notation "$r_p$" is chosen to represent Pearson correlation coefficients, and the notation "$r_s$" is chosen to represent Spearman's rank correlation coefficients. In Tables 5(a) and 5(b), the notations "$r_{p1}$",
"$r_{p2}$", "$r_{p3}$" and "$r_{p4}$" denote the Pearson correlation coefficients for mono layers, double layers, triple layers, and quadruple layers respectively. In a similar manner, the Spearman's rank correlation coefficient notations are "$r_{s1}$", "$r_{s2}$", "$r_{s3}$" and "$r_{s4}$".

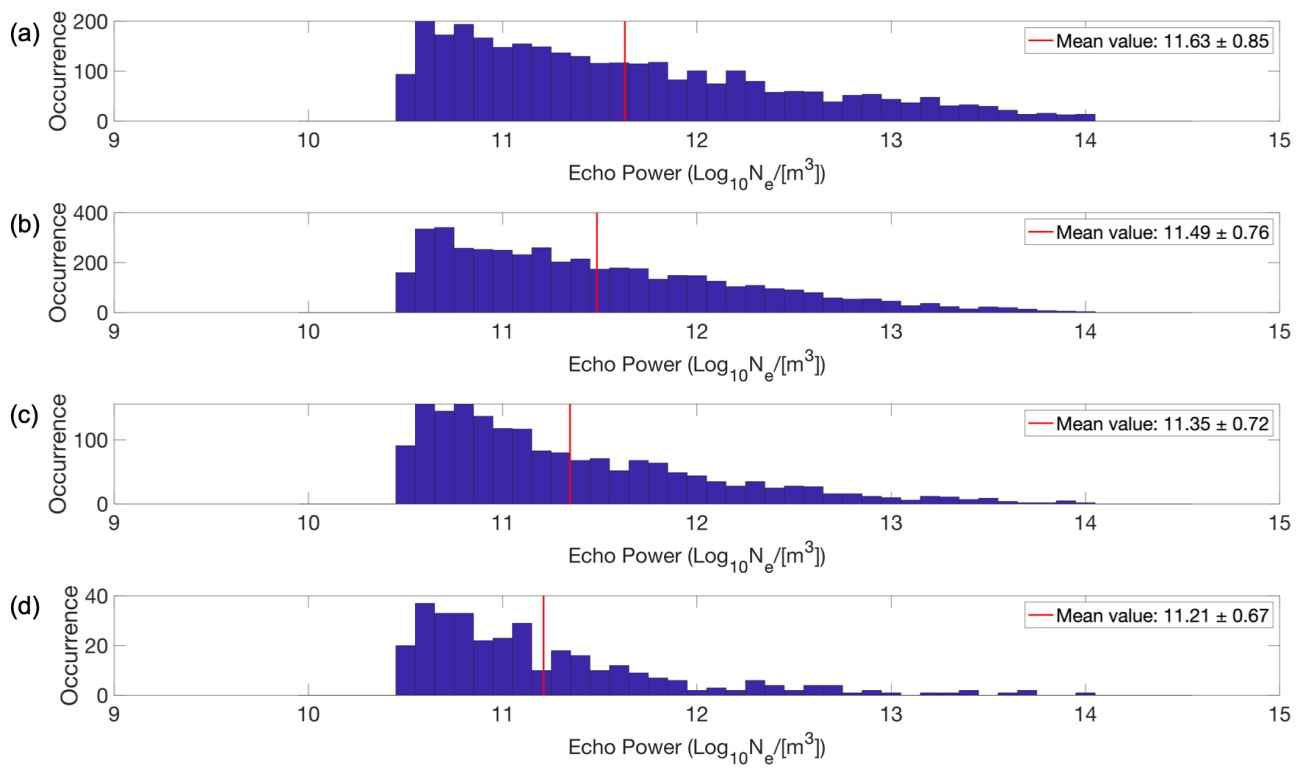

**Figure 10.** Echo power in the PMSE during solar maximum for **(a)** mono layers, **(b)** multi layers with 2 layers, **(c)** multi layers with 3 layers, and **(d)** multi layers with 4 layers. Each subplot has its respective mean echo power represented with a red line on the graph, and specified in the legend together with one standard deviation.

**Table 4.** Pearson and Spearman's rank correlation coefficients for all layers together, for solar maximum and solar minimum.

<table>
<tr><td></td><td></td><td colspan="4">Solar minimum</td></tr>
<tr><td></td><td></td><td>Electron density</td><td>Echo power</td><td>Thickness</td><td>Altitude</td></tr>
<tr><td rowspan="8">Solar maximum</td><td rowspan="2">Electron density</td><td></td><td>$r_p = 0.213$</td><td>$r_p = 0.251$</td><td>$r_p = -0.079$</td></tr>
<tr><td></td><td>$r_s = 0.163$</td><td>$r_s = 0.232$</td><td>$r_s = -0.058$</td></tr>
<tr><td rowspan="2">Echo power</td><td>$r_p = 0.338$</td><td></td><td>$r_p = 0.521$</td><td>$r_p = -0.165$</td></tr>
<tr><td>$r_s = 0.305$</td><td></td><td>$r_s = 0.631$</td><td>$r_s = -0.162$</td></tr>
<tr><td rowspan="2">Thickness</td><td>$r_p = 0.480$</td><td>$r_p = 0.510$</td><td></td><td>$r_p = -0.153$</td></tr>
<tr><td>$r_s = 0.392$</td><td>$r_s = 0.631$</td><td></td><td>$r_s = -0.169$</td></tr>
<tr><td rowspan="2">Altitude</td><td>$r_p = 0.011$</td><td>$r_p = -0.034$</td><td>$r_p = 0.039$</td><td></td></tr>
<tr><td>$r_s = 0.003$</td><td>$r_s = -0.031$</td><td>$r_s = 0.024$</td><td></td></tr>
</table>

**Table 5.** **(a)** Pearson correlation coefficients for mono and multi layers separately, for solar maximum and solar minimum. **(b)** Spearman's rank correlation coefficients for mono and multi layers separately, for solar maximum and solar minimum.

| (a) | | Solar minimum | | | |
|---|---|---|---|---|---|
| | | Electron density | Echo power | Thickness | Altitude |
| Solar maximum | Electron density | | $r_{p1} = 0.270$ $r_{p2} = 0.247$ $r_{p3} = 0.163$ $r_{p4} = 0.199$ | $r_{p1} = 0.376$ $r_{p2} = 0.273$ $r_{p3} = 0.226$ $r_{p4} = 0.168$ | $r_{p1} = -0.339$ $r_{p2} = 0.010$ $r_{p3} = 0.048$ $r_{p4} = 0.054$ |
| | Echo power | $r_{p1} = 0.501$ $r_{p2} = 0.259$ $r_{p3} = 0.224$ $r_{p4} = 0.306$ | | $r_{p1} = 0.455$ $r_{p2} = 0.574$ $r_{p3} = 0.608$ $r_{p4} = 0.514$ | $r_{p1} = -0.071$ $r_{p2} = -0.186$ $r_{p3} = -0.228$ $r_{p4} = -0.210$ |
| | Thickness | $r_{p1} = 0.695$ $r_{p2} = 0.393$ $r_{p3} = 0.246$ $r_{p4} = 0.264$ | $r_{p1} = 0.534$ $r_{p2} = 0.482$ $r_{p3} = 0.508$ $r_{p4} = 0.541$ | | $r_{p1} = -0.110$ $r_{p2} = -0.199$ $r_{p3} = -0.167$ $r_{p4} = -0.161$ |
| | Altitude | $r_{p1} = 0.091$ $r_{p2} = -0.079$ $r_{p3} = -0.046$ $r_{p4} = 0.030$ | $r_{p1} = 0.087$ $r_{p2} = -0.052$ $r_{p3} = -0.118$ $r_{p4} = -0.184$ | $r_{p1} = 0.131$ $r_{p2} = 0.031$ $r_{p3} = -0.040$ $r_{p4} = -0.113$ | |

| (b) | | Solar minimum | | | |
|---|---|---|---|---|---|
| | | Electron density | Echo power | Thickness | Altitude |
| Solar maximum | Electron density | | $r_{s1} = 0.245$ $r_{s2} = 0.179$ $r_{s3} = 0.178$ $r_{s4} = 0.123$ | $r_{s1} = 0.428$ $r_{s2} = 0.215$ $r_{s3} = 0.178$ $r_{s4} = 0.173$ | $r_{s1} = -0.292$ $r_{s2} = 0.006$ $r_{s3} = 0.045$ $r_{s4} = 0.047$ |
| | Echo power | $r_{s1} = 0.494$ $r_{s2} = 0.239$ $r_{s3} = 0.202$ $r_{s4} = 0.232$ | | $r_{s1} = 0.603$ $r_{s2} = 0.643$ $r_{s3} = 0.635$ $r_{s4} = 0.542$ | $r_{s1} = -0.047$ $r_{s2} = -0.188$ $r_{s3} = -0.240$ $r_{s4} = -0.208$ |
| | Thickness | $r_{s1} = 0.668$ $r_{s2} = 0.311$ $r_{s3} = 0.202$ $r_{s4} = 0.230$ | $r_{s1} = 0.615$ $r_{s2} = 0.621$ $r_{s3} = 0.637$ $r_{s4} = 0.595$ | | $r_{s1} = -0.168$ $r_{s2} = -0.185$ $r_{s3} = -0.141$ $r_{s4} = -0.124$ |
| | Altitude | $r_{s1} = 0.095$ $r_{s2} = -0.052$ $r_{s3} = -0.031$ $r_{s4} = 0.058$ | $r_{s1} = 0.111$ $r_{s2} = -0.051$ $r_{s3} = -0.107$ $r_{s4} = -0.190$ | $r_{s1} = 0.161$ $r_{s2} = 0.008$ $r_{s3} = -0.052$ $r_{s4} = -0.076$ | |

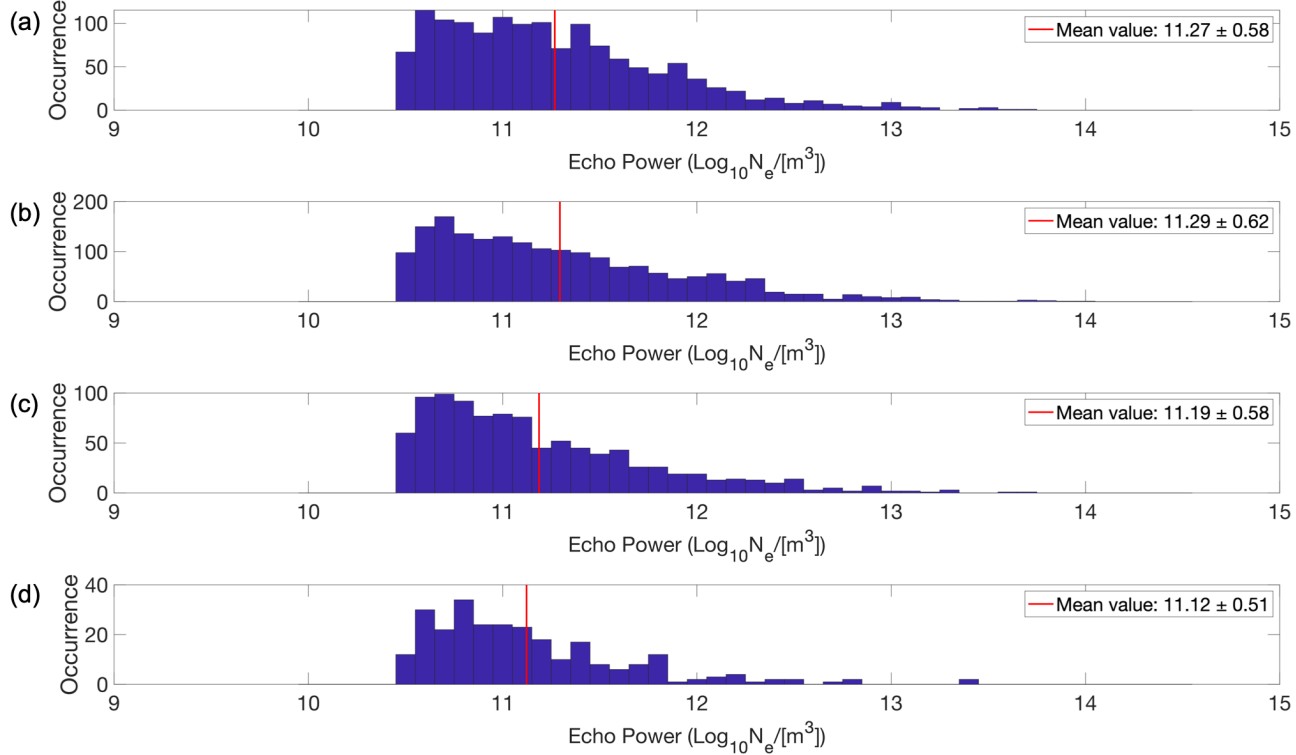

**Figure 11.** Echo power in the PMSE during solar minimum for **(a)** mono layers, **(b)** multi layers with 2 layers, **(c)** multi layers with 3 layers, and **(d)** multi layers with 4 layers. Each subplot has its respective mean echo power represented with a red line on the graph, and specified in the legend together with one standard deviation.

In Table 4, it is observed that the electron density at 92 km altitude and the echo power are positively correlated with the thickness of all the layers for both solar maximum and solar minimum. This is also the case for Tables 5(a), and 5(b). During solar maximum, the positive correlation between electron density and thickness is greater than during solar minimum, but this is not observed between echo power and thickness. In Tables 4, the Pearson correlation coefficient of 0.480 for solar maximum suggests a moderate positive linear relationship between electron density and thickness, while the Spearman's rank correlation coefficient of 0.392 indicates a moderate positive monotonic relationship between the variables for the same case. Since the two values are similar, it suggests that during solar maximum there is a consistent association between electron density and thickness. In Tables 5(a), and 5(b), we observe that the Pearson correlation coefficient and Spearman's rank correlation coefficient between electron density and thickness decrease as the number of multi layers increases. Specifically, in both cases the highest correlation is observed for solar maximum and mono layers, with a Pearson coefficient of 0.695 and a Spearman's rank coefficient of 0.668. This could possibly indicate that at higher ionization levels at this altitude, the PMSE mono layers are thicker. Conversely, the lowest correlations were obtained for solar minimum and the largest number of multi layers, which is 4, with a Pearson coefficient of 0.168 and a Spearman's rank coefficient of 0.173.

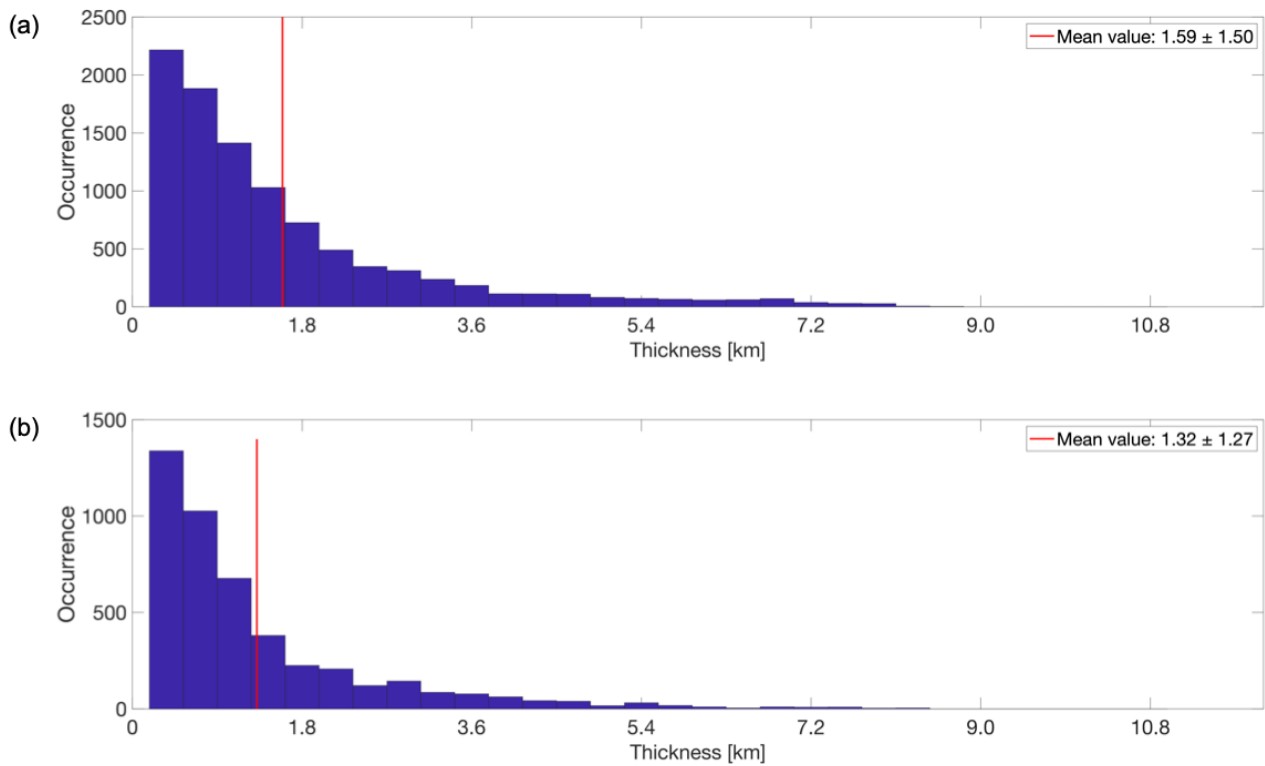

**Figure 12.** Thickness distribution of the layers for all layers combined during **(a)** solar maximum and **(b)** solar minimum. Each subplot was its respective mean thickness represented with a red line on the graph, and specified in the legend together with one standard deviation.

From Tables 4, 5(a), and 5(b) we notice a weak negative correlation between the echo power in the PMSE and altitude for all layers during both solar maximum and solar minimum. The strongest negative correlation is found for 3 multi layers, with a Pearson coefficient of -0.228 and a Spearman's rank coefficient of -0.240. Notably, altitude appears to be uncorrelated with the other variables, implying that additional factors may be influencing the formation of PMSE at specific altitudes. For example, this could be attributed to mesopause conditions, gravity wave wavelength and ice particle size.

From Tables 4, 5(a), and 5(b) we notice overall the positive correlation between the electron density at 92 km altitude and the echo power in the PMSE for all the layers and for both solar maximum and solar minimum. For Tables 5(a) and 5(b), we note that the highest Pearson correlation coefficient and Spearman's rank correlation coefficient are obtained for mono layers. Specifically for solar maximum, the Pearson coefficient is 0.501 and the Spearman's rank coefficient is 0.494, while for solar minimum, the Pearson coefficient is 0.270 and the Spearman's rank coefficient is 0.245. These results can possibly suggest that

at higher ionization levels at 92 km altitude, the PMSE have a higher intensity, indicated by a higher echo power, particularly in the case of mono layers during solar maximum. On the other hand, the lowest correlations were found for multi layers containing three layers, with a Pearson coefficient of 0.224 and a Spearman's rank coefficient of 0.202 for solar maximum and a Pearson coefficient of 0.306 and a Spearman's rank coefficient of 0.232 for solar minimum.

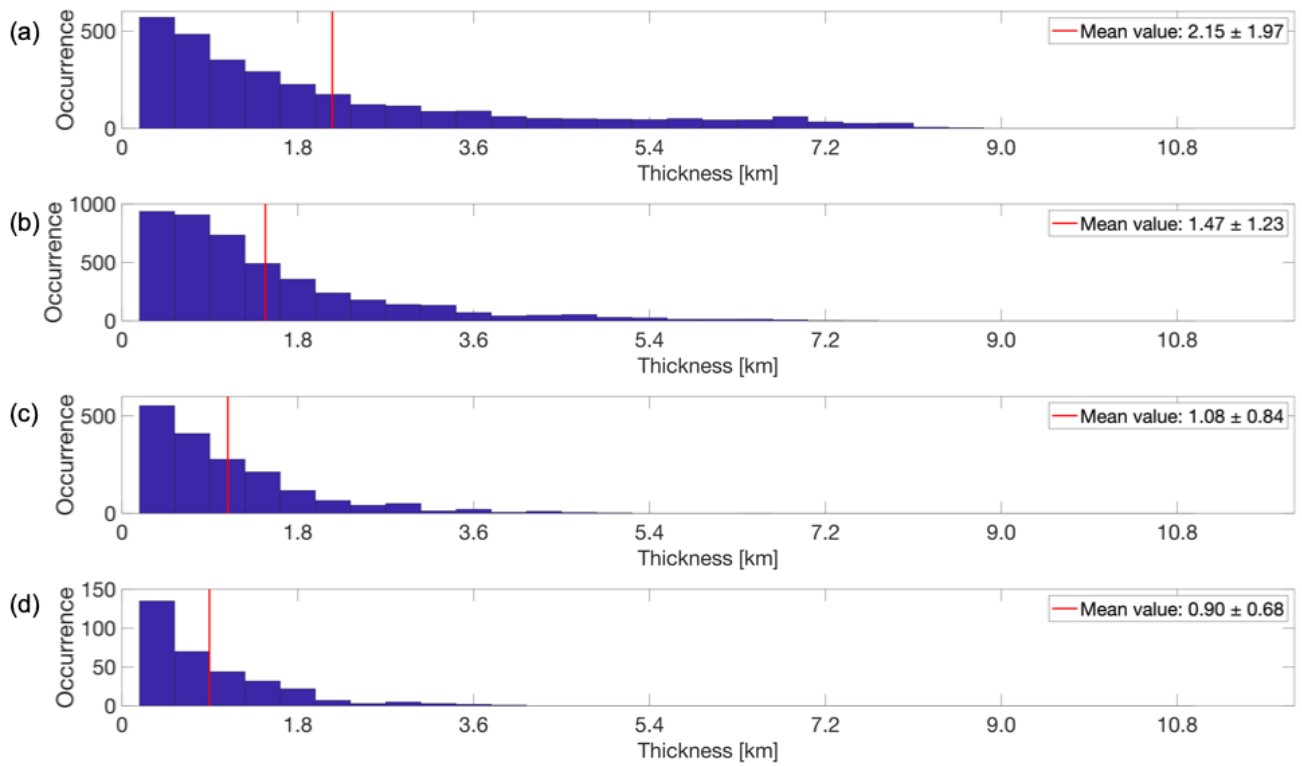

**Figure 13.** Thickness distribution during solar maximum for **(a)** mono layers, **(b)** multi layers with 2 layers, **(c)** multi layers with 3 layers, and **(d)** multi layers with 4 layers. Each subplot was its respective mean thickness represented with a red line on the graph, and specified in the legend together with one standard deviation.

Narayanan et al. (2022) investigated the effects of particle precipitation on PMSE formation using electron densities from 90 to 95 km. They found a clear response in the power of the PMSE echoes during particle precipitation events: in all their cases, an increase in PMSE power was observed in association with particle precipitations. However, Narayanan et al. (2022) say that the particle precipitation does not seem to be related to the very existence of PMSE, and that there seem to be no linear relationship between both, which is consistent with the results of our study. Specifically, we observe weak Pearson correlation coefficients during the solar minimum, as reported in Table 5(a), consistent with the findings of Narayanan et al. (2022) who analyzed EISCAT VHF observations from 2019, a period corresponding to the solar minimum. However, our results indicate slightly higher Pearson correlation coefficients during solar maximum, particularly for mono layers. It would be worthwhile to conduct a similar investigation as Narayanan et al. (2022) during the solar maximum phase of a solar cycle. These findings should be interpreted with care, considering that our study differs from that of Narayanan et al. (2022) in several ways. Specifically, our data selection process did not require the simultaneous presence of PMSE and particle precipitation.

From Table 4, one can notice that for the combination of echo power and electron density during solar maximum, the obtained Pearson correlation coefficient is 0.338 and the Spearman's rank correlation coefficient is 0.305. In their study, Rauf

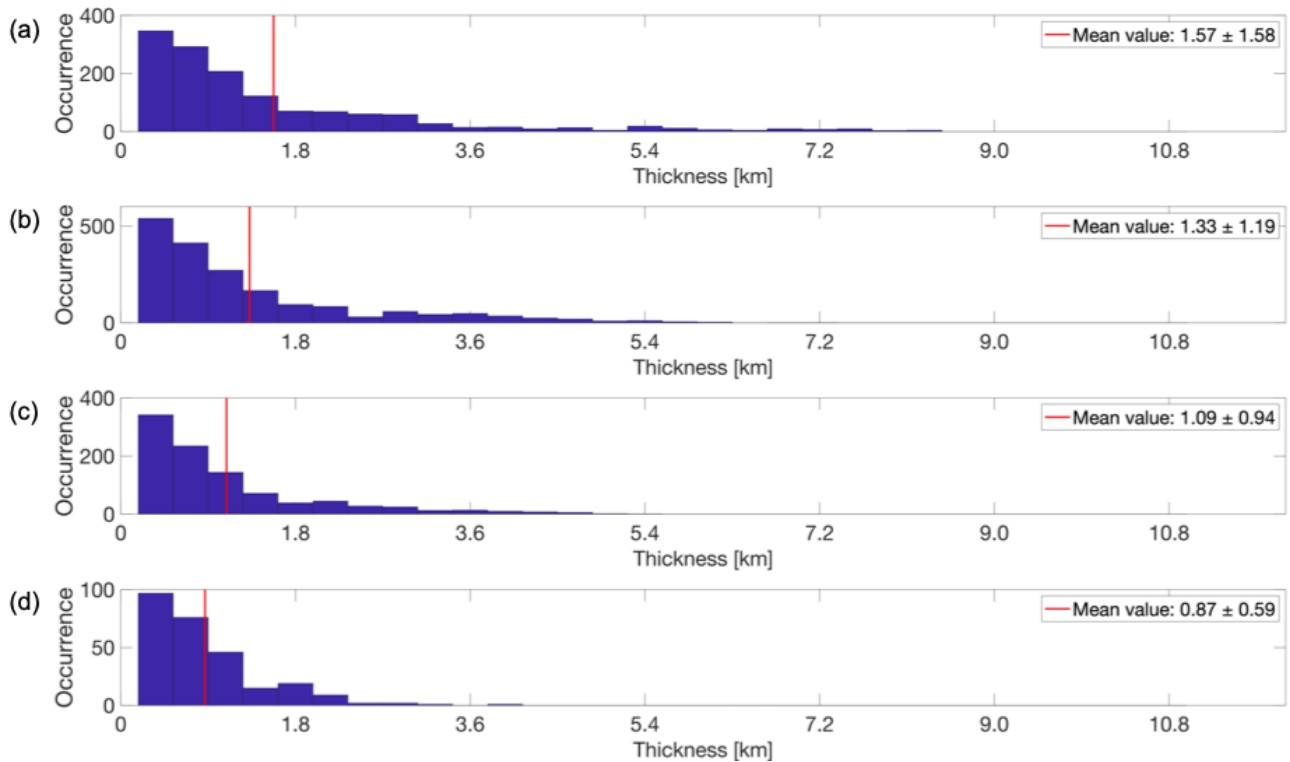

**Figure 14.** Thickness distribution during solar minimum for **(a)** mono layers, **(b)** multi layers with 2 layers, **(c)** multi layers with 3 layers, and **(d)** multi layers with 4 layers. Each subplot was its respective mean thickness represented with a red line on the graph, and specified in the legend together with one standard deviation.

et al. (2018a) used EISCAT VHF data to investigate the correlation between PMSE strength and particle precipitation, over a dataset of 111 hours, or 5 days of observation. However in their case, they derived the Pearson and Spearman correlation coefficients between their PMSE proxy which is equivalent to our use of the term "echo power", and the electron density at 90 km altitude instead of 92 km for us. Nevertheless, it is interesting to note that they also found a positive correlation between echo power and electron density with 0.15 for the Pearson correlation coefficient, and 0.24 for the Spearman correlation coefficient. It is important to note that during their analysis, Rauf et al. (2018a) only selected data from 8 to 12 July 2013, when PMSE and particle precipitation were occurring simultaneously. In our study, we included data from the year 2013 in the solar maximum period. Hence, we compare the correlation coefficients from Rauf et al. (2018a) with our own coefficients for the solar maximum. While both studies discovered a positive correlation, our findings had higher correlation coefficients than Rauf et al. (2018a) study. One factor which could explain this difference might be the fact that in Rauf et al. (2018a) data, PMSE and particle precipitation was always occurring simultaneously, while in our analysis, data was selected solely based on the presence of PMSE, without any filtering based on the occurrence of particle precipitation. It should be noted that while PMSE was present in all of our cases, there may have been instances where particle precipitation was present and instances

where it was not. Another factor might be that we used a lower threshold for PMSE detection than Rauf et al. (2018a), due to the fact that we used a classification model on the data before hand. We used the threshold $N_e > 3.2 \times 10^{10} m^{-3}$ while Rauf et al. (2018a) used $N_e > 4.6 \times 10^{11} m^{-3}$.

## 4 Conclusions

The altitude, the echo power and the thickness of layers in PMSE have on average higher values during solar maximum than
during solar minimum. During the PMSE occurrence, as expected, the electron density at 92 km is on average higher during solar maximum than solar minimum. Taking into account the findings presented by Lübken et al. (2021) that show an increase in ice particle size over time in conjunction with these results, it is difficult to isolate the exact mechanisms by which the PMSE properties are affected. None the less, breaking down the multilayer sets into individual layers reveals a consistent trend: in both solar maximum and solar minimum cases, the altitude of the top layer tends to rise with an increasing number of
multilayers. This tendency extends to the second and third highest layers as well. Our findings support the conclusions drawn by Hoffmann et al. (2005) regarding the altitude and occurrence rate of both mono and multiple layers. Additionally, when examining the lowest layer in various multilayer sets, the lowest layer almost always aligns with the NLC altitude as reported by Fiedler et al. (2003) of 83.3 km. The recent work by Vellalassery et al. (2024) addresses the variation of NLCs throughout the solar cycle. They used the Leibniz Institute Middle Atmosphere (LIMA) model and the Mesospheric Ice Microphysics
and Transport (MIMAS) model over the years 1849 to 2019, corresponding to 15 solar cycles. Their findings indicate that NLC altitudes increase during periods of solar maximum and decrease during solar minimum. Additionally, they observed a long-term decline in NLC altitude, attributed to the overall shrinking of the atmosphere. Our findings align with those results, as we observed a lower altitude of the PMSE during the solar minimum period (years 2019 and 2020) compared to the solar maximum phase (years 2013 to 2015).

We have observed that the thickness of the layers decreases as the number of multi-layers increases, indicating that a single mono-layer will be thicker than the separate layers of a set of two multi-layers, which in turn will be thicker than the separate layers of three multi-layers, and so on. This is mostly the case for layers one to three and for both solar maximum and solar minimum. Furthermore, the echo power was found to decrease with increasing multi layers, but only in the case of solar maximum, and mostly for layers one to three. This suggests that there may be a relationship between the number of layers,
echo power, and thickness. Our study is consistent with the findings of Li et al. (2016) where they found that the thickness of multi layers decreases with increasing number of multilayers.

Based on our investigation, we have found that the electron density at 92 km altitude and the echo power are positively correlated with the thickness for all the layers and for both solar maximum and solar minimum, except for four multi layers at solar minimum. We also found similar results as Rauf et al. (2018a), discovering a positive correlation between electron density
and echo power, especially for mono layers and during solar maximum. This can possibly suggest that under those conditions and at higher ionization levels at 92 km altitude, the PMSE are stronger, indicated by a higher echo power. The electron density was highly correlated with the thickness of the layers, except for solar minimum and 4 multilayers. The correlation is the

strongest especially for solar maximum and mono layers, which indicates that at higher ionization levels at 92km altitude, the PMSE mono layers are commonly thicker. Comparing our results with Li et al. (2016) led us to hypothesize that the thickness of the layers could be related to the vertical wavelength of gravity waves, with larger wavelengths producing thicker layers. Further investigations could explore this hypothesis, potentially providing a means to infer the wavelength of gravity waves through PMSE observations at these altitudes.

For both solar maximum and solar minimum periods, the mono layers attained the lowest average electron density of their respective seasons, though the trend was relatively weak. An argument could be made that higher electron densities at ionospheric altitudes might be necessary to generate multi-layered PMSEs, though this requires more investigation.

A parallel can be drawn with the findings of Schäfer et al. (2020) regarding multi layered NLC, where both our studies found a similar occurrence rate for thick layer formation above 1 km thickness. In light of the similarities in multi-layer formation between PMSE and NLC, future studies may be able to utilize findings from NLC research to gain insights into PMSE dynamics.

In conclusion, the mechanism of the formation PMSE might be presently well understood, however the exact conditions leading to multi-layered PMSE formation remain unclear, and further investigation is required. Hoffmann et al. (2005) proposed that PMSE layering can be explained by the stratification of ice particles resulting from successive nucleation cycles near the mesopause, followed by growth and sedimentation. Other authors hypothesized a potential connection between PMSE multilayers and gravity waves (Li et al., 2016), (Hoffmann et al., 2005). Our hypothesis on the formation of multi-layered PMSE is that gravity waves transport particles into regions of low temperature at varying altitudes. In these conditions, ice particles can form and grow. This process may impact the size of ice particles, which in turn could affect their spatial distribution via sedimentation, and potentially influencing the formation of multilayers. Therefore, for example, future research could include further investigation of the connections between multi-layered PMSE formation, winds and gravity waves. One possible way to do this is to measure gravity waves using the EISCAT radar, (Günzkofer et al., 2023). Utilizing the dissipative anelastic gravity wave dispersion relation, Günzkofer et al. (2023) derive vertical wind profiles within the lower thermosphere. This is a promising avenue for further measuring of gravity waves during PMSE occurrences. Understanding the complex interplay of the factors involving the formation of PMSE is crucial to gain insights into the thermodynamic and fluid dynamic processes occurring at altitudes between 80 to 90 km. While differences between the results from observations during solar maximum and during solar minimum considering all the layers together are statistically significant, the cause for the differences needs to be confirmed by future studies.

*Data availability.* EISCAT VHF data are available under https://madrigal.eiscat.se/madrigal/ (accessed on 15 January 2023).

*Author contributions.* Conceptualization, D.J., P.S., D.H. and I.M.; Data curation, D.J.; Funding acquisition, I.M.; Investigation, D.J.; Project administration, I.M.; Software, D.J.; Supervision, P.S., D.H. and I.M.; Validation, P.S., D.H. and I.M.; Writing—original draft, D.J.; Writing—review and editing, D.J., P.S. D.H. and I.M. All authors have read and agreed to the published version of the manuscript.

*Competing interests.* At least one of the (co-)authors is a member of the editorial board of Annales Geophysicae. The peer-review procedure was conducted by an impartial editor, and the authors declare that they have no additional competing interests. The funders had no role in the design of the study; in the collection, analyses, or interpretation of data; in the writing of the manuscript, or in the decision to publish the results.

*Acknowledgements.* This work was carried out within a project funded by Research Council of Norway, NFR 275503. The Norwegian
participation in EISCAT and EISCAT3D is funded by Research Council of Norway, through research infrastructure grant 245683. The EISCAT International Association is supported by research organizations in Norway (NFR), Sweden (VR), Finland (SA), Japan (NIPR and STEL), China (CRIPR), and the United Kingdom (NERC).

Devin Huyghebaert was funded during this study through a UiT The Arctic University of Norway contribution to the EISCAT_3D project funded by Research Council of Norway through research infrastructure grant 245683.

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

## Appendix A: Figures

## Appendix B: Tables

**Table B1.** P-values for all combinations of layers shown in Fig. 4 and Fig. 5.

| P-Values | | Solar Minimum | | | | | | | | | |
|---|---|---|---|---|---|---|---|---|---|---|---|
| | | Mono Layers | Layers 1 of 2 | Layers 2 of 2 | Layers 1 of 3 | Layers 2 of 3 | Layers 3 of 3 | Layers 1 of 4 | Layers 2 of 4 | Layers 3 of 4 | Layers 4 of 4 |
| Solar Maximum | Mono Layers | | P <0.0001 | P <0.0001 | P <0.0001 | 0.3618 | P <0.0001 | P <0.0001 | P <0.0001 | 0.0027 | P <0.0001 |
| | Layers 1 of 2 | P <0.0001 | | P <0.0001 | P <0.0001 | P <0.0001 | P <0.0001 | P <0.0001 | 0.0268 | P <0.0001 | P <0.0001 |
| | Layers 2 of 2 | P <0.0001 | P <0.0001 | | P <0.0001 | P <0.0001 | P <0.0001 | P <0.0001 | P <0.0001 | P <0.0001 | P <0.0001 |
| | Layers 1 of 3 | P <0.0001 | P <0.0001 | P <0.0001 | | P <0.0001 | P <0.0001 | 0.0106 | P <0.0001 | P <0.0001 | P <0.0001 |
| | Layers 2 of 3 | P <0.0001 | P <0.0001 | P <0.0001 | P <0.0001 | | P <0.0001 | P <0.0001 | P <0.0001 | P <0.0001 | P <0.0001 |
| | Layers 3 of 3 | P <0.0001 | P <0.0001 | 0.0002 | P <0.0001 | P <0.0001 | | P <0.0001 | P <0.0001 | P <0.0001 | 0.0001 |
| | Layers 1 of 4 | P <0.0001 | P <0.0001 | P <0.0001 | 0.0001 | P <0.0001 | P <0.0001 | | P <0.0001 | P <0.0001 | P <0.0001 |
| | Layers 2 of 4 | P <0.0001 | 0.0448 | P <0.0001 | P <0.0001 | P <0.0001 | P <0.0001 | P <0.0001 | | P <0.0001 | P <0.0001 |
| | Layers 3 of 4 | 0.0411 | P <0.0001 | P <0.0001 | P <0.0001 | P <0.0001 | P <0.0001 | P <0.0001 | P <0.0001 | | P <0.0001 |
| | Layers 4 of 4 | P <0.0001 | P <0.0001 | P <0.0001 | P <0.0001 | P <0.0001 | P <0.0001 | P <0.0001 | P <0.0001 | P <0.0001 | |

**Table B2.** P-values for all combinations of layers and parameters shown in Fig. 3, Fig. A1, Fig. A2, Fig. 6, Fig. 7, Fig. 8, Fig. 9, Fig. 10, Fig. 11, Fig. 12, Fig. 13, and Fig. 14.

| | | Altitude | Electron Density | Echo Power | Thickness |
|---|---|---|---|---|---|
| Solar Maximum | Layers 1-2 | P = 0.6462 | P <0.0001 | P <0.0001 | P <0.0001 |
| | Layers 1-3 | P <0.0001 | P = 0.0003 | P <0.0001 | P <0.0001 |
| | Layers 1-4 | P = 0.0002 | P = 0.0831 | P <0.0001 | P <0.0001 |
| | Layers 2-3 | P <0.0001 | P = 0.0804 | P <0.0001 | P <0.0001 |
| | Layers 2-4 | P = 0.0014 | P = 0.4000 | P <0.0001 | P <0.0001 |
| | Layers 3-4 | P = 0.8035 | P = 1.0000 | P = 0.0012 | P = 0.0002 |
| Solar Minimum | Layers 1-2 | P = 0.6808 | P <0.0001 | P = 0.3483 | P <0.0001 |
| | Layers 1-3 | P = 0.1098 | P <0.0001 | P = 0.0009 | P <0.0001 |
| | Layers 1-4 | P = 0.3030 | P <0.0001 | P = 0.0001 | P <0.0001 |
| | Layers 2-3 | P = 0.0481 | P <0.0001 | P <0.0001 | P <0.0001 |
| | Layers 2-4 | P = 0.2284 | P = 0.0091 | P <0.0001 | P <0.0001 |
| | Layers 3-4 | P = 1.0000 | P = 0.5707 | P = 0.0728 | P = 0.0002 |
| **Sol Max. - Min.** | | P <0.0001 | P <0.0001 | P <0.0001 | P <0.0001 |

**Table B3.** P-values for the correlation coefficients for all layers together during solar maximum and solar minimum shown in Table 4.

| P-Values | | Solar minimum | | | |
|---|---|---|---|---|---|
| | | Electron density | Echo power | Thickness | Altitude |
| Solar Max. | Electron density | | 1.53E-27 | 1.38E-54 | 1.06E-04 |
| | Echo power | 1.02E-203 | | 0 | 2.51E-28 |
| | Thickness | 0 | 0 | | 1.94E-30 |
| | Altitude | 0.772 | 2.24E-03 | 0.0175 | |

**Table B4.** P-values for the correlation coefficients for the mono and multi layers separately, during solar maximum and solar minimum shown in Table 5 b).

| P-Values | | Solar minimum | | | |
|---|---|---|---|---|---|
| | | Electron density | Echo power | Thickness | Altitude |
| Solar Maximum | Electron density | | Layer1 = 1.86E-19<br>Layer2 = 1.49E-14<br>Layer3 = 4.17E-04<br>Layer4 = 0.0489 | Layer1 = 1.02E-59<br>Layer2 = 2.84E-08<br>Layer3 = 4.17E-04<br>Layer4 = 0.00542 | Layer1 = 2.08E-27<br>Layer2 = 0.800<br>Layer3 = 0.165<br>Layer4 = 0.455 |
| | Echo power | Layer1 = 4.06E-183<br>Layer2 = 5.68E-58<br>Layer3 = 4.29E-12<br>Layer4 = 3.19E-05 | | Layer1 = 3.58E-139<br>Layer2 = 5.62E-112<br>Layer3 = 4.17E-04<br>Layer4 = 6.96E-22 | Layer1 = 0.0760<br>Layer2 = 2.30E-16<br>Layer3 = 2.51E-14<br>Layer4 = 5.92E-04 |
| | Thickness | Layer1 = 0<br>Layer2 = 9.23E-99<br>Layer3 = 1.65E-17<br>Layer4 = 3.60E-05 | Layer1 = 4.186E-319<br>Layer2 = 0<br>Layer3 = 8.51E-205<br>Layer4 = 1.89E-32 | | Layer1 = 2.87E-10<br>Layer2 = 8.82E-06<br>Layer3 = 4.17E-04<br>Layer4 = 0.0418 |
| | Altitude | Layer1 = 1.80E-07<br>Layer2 = 5.19E-04<br>Layer3 = 0.194<br>Layer4 = 0.305 | Layer1 = 6.87E-10<br>Layer2 = 5.85E-04<br>Layer3 = 5.32E-06<br>Layer4 = 6.02E-04 | Layer1 = 2.93E-19<br>Layer2 = 0.592<br>Layer3 = 0.0288<br>Layer4 = 0.174 | |

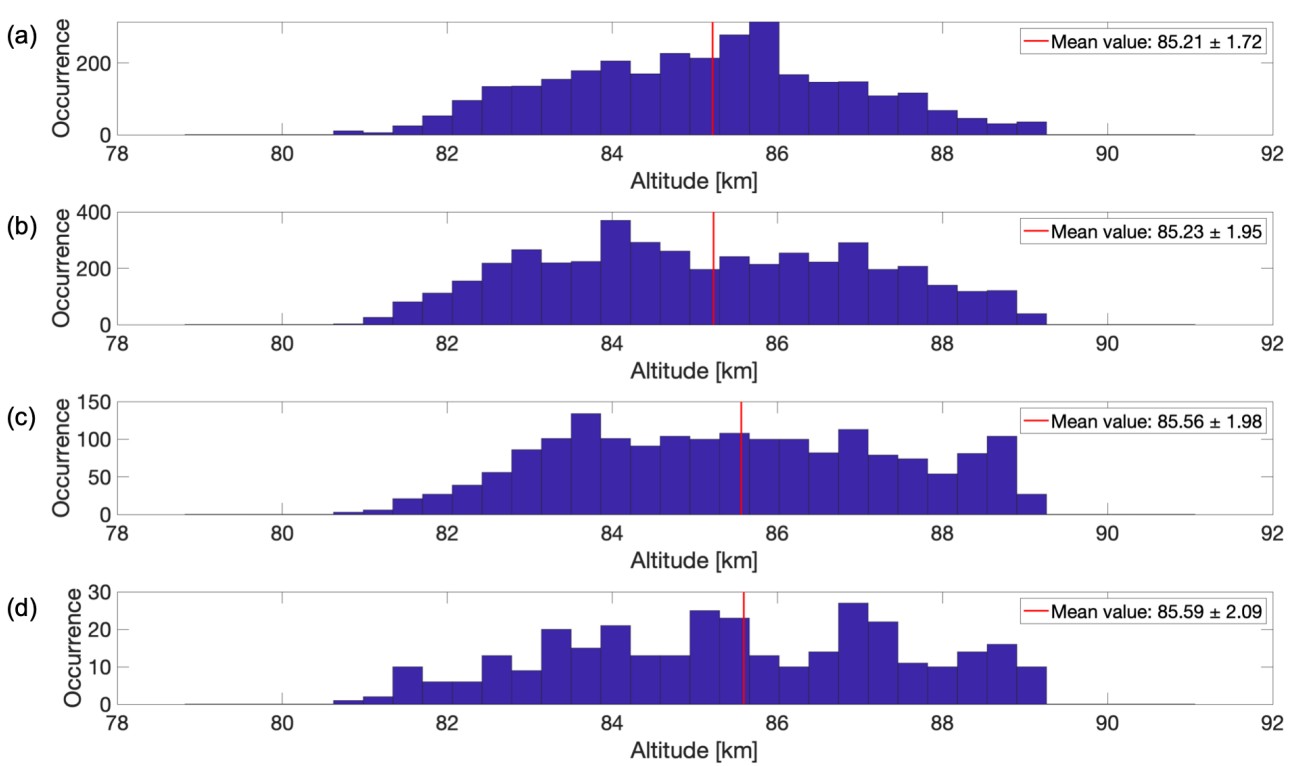

**Figure A1.** Altitude distribution of the data during solar maximum for **(a)** mono layers, **(b)** multi layers with 2 layers, **(c)** multi layers with 3 layers, and **(d)** multi layers with 4 layers. Each subplot was its respective averaged mean altitude of all the multilayers, represented with a red line on the graph, and specified in the legend together with one standard deviation.

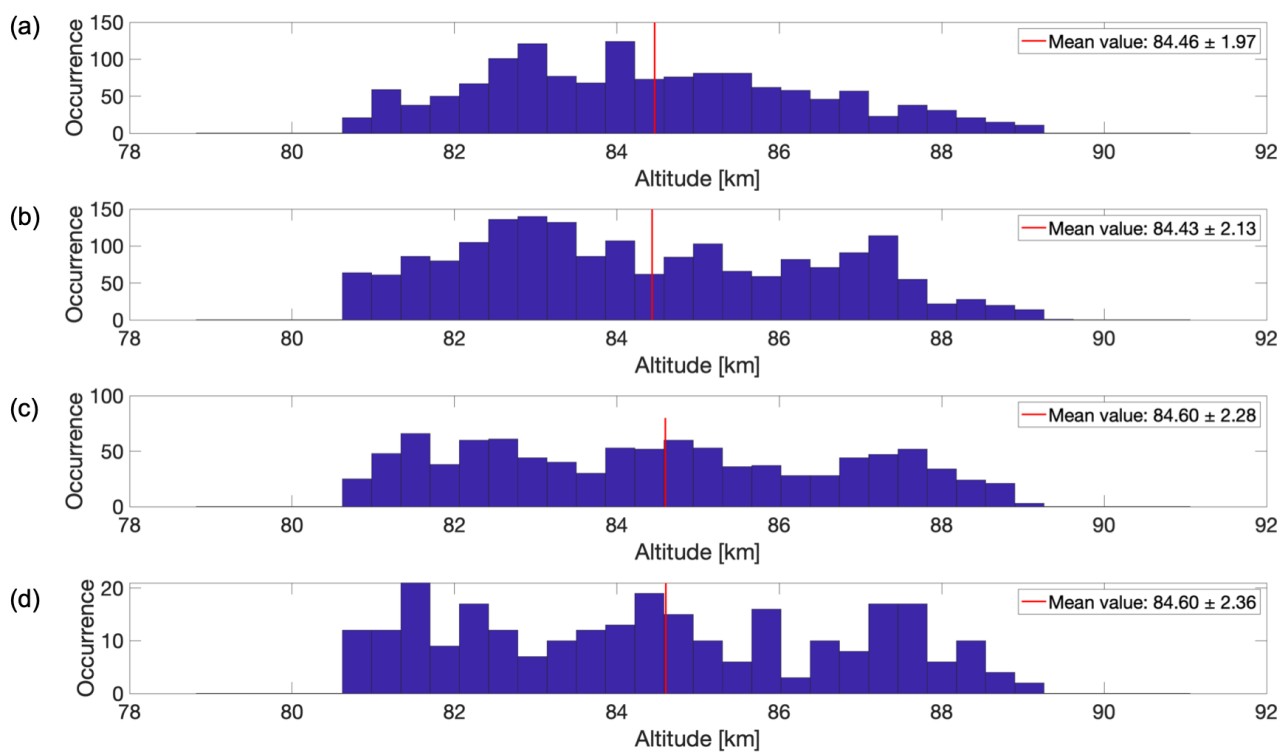

**Figure A2.** Altitude distribution of the data during solar minimum for **(a)** mono layers, **(b)** multi layers with 2 layers, **(c)** multi layers with 3 layers, and **(d)** multi layers with 4 layers. Each subplot was its respective averaged mean altitude of all the multilayers, represented with a red line on the graph, and specified in the legend together with one standard deviation.