# Peer review of "Polar Mesospheric Summer Echo (PMSE) Multilayer Properties During Solar Maximum and Solar Minimum"

_EGUsphere, 2023_

## Author Comment (AC1)

Response to Reviewer 1:

We thank the reviewer for the constructive and helpful comments, which helped us to improve the manuscript. We took all comments into account when revising the manuscript. In the text below we describe the modifications and list our responses together with the reviewer's comments that are repeated here in blue color. Our answers are given in black.

This a quite interesting study of PMSE layers and should be published in AnGeo after suitable corrections are made. I have very little questions concerning the observations. This was done quite well. However I do have some comments/questions about the conclusions and implied physical causes.

**Comments**

1- Title. It would be best to spell out PMSE for the readers of AnGeo. I suspect that many of the readership will not know what that is. Also since you have a strong focus on multilayers, perhaps a better title would be "Polar Mesospheric Summer Echo (PMSE) Multilayer Properties During Solar Maximum and Solar Minimum"? Something like that?

We agree that it would make sense to emphasize the multilayer aspect of the study in the title, and to spell out the acronym "PMSE". The title was modified accordingly to the reviewer's comment in the following way:

«Polar Mesospheric Summer Echo (PMSE) Multilayer Properties During Solar Maximum and Solar Minimum»

2- Since you have invoked particle precipitation as a possible source for the radar wave scattering, you should mention that the magnetic latitude of Tromso. I believe it is 67 degrees. The geographic latitude is 69 deg, but for particle precipitation, the magnetic latitude is more important. The "auroral zone" is 60 to 70 deg magnetic latitude, so your observations have been taken in the center of it. During relatively quiet intervals the auroral precipitation occurs in the auroral zone. During solar maximum and during storms the precipitation is at slightly lower magnetic latitudes, as low as 55 deg. The greatest auroral precipitation is now known to occur in the declining phase of the solar cycle associated with high speed solar wind streams. A reference to this can be found in JGR, 100, A11, 21717, 1995. This precipitation occurs in the auroral zone. I think what you have stated about precipitation during solar max and solar min is okay as is. However I point out some of the subtleties for your information.

Thank you for sharing this very interesting information and providing a deeper understanding of the relationship between the phases of the solar cycle and auroral precipitation. We have incorporated additional details in the manuscript with the following sentence:

«The geographical coordinates of the EISCAT VHF radar are 69°35′N and 19°14′E; its geomagnetic latitude and longitude are respectively 66.73° and 102.18°.»

3- During solar maximum there is more EUV radiation. Depending on season, perhaps that can be a cause for greater electron densities during solar maximum? Also could solar heating expand the atmosphere slightly giving you your height difference during solar maximum? Please discuss in the body of the text.

The solar maximum phase is characterized by an increased number of sunspots and higher levels of EUV radiation compared to the solar minimum phase. This EUV radiation has impacts on various parameters in the atmosphere, including temperature (as mentioned by the reviewer), water vapor content, electron density, maybe also atmospheric circulation, and gravity wave activity. Given all these parameters, it is challenging to determine with certainty how the mesosphere would behave during solar maximum when there is a greater amount of EUV radiation. We now have included the sunspot number and the F10.7 index in Table 1. Here is the new Table 1 :

| | F10.7 cm Flux | Sunspot Number | Year | Date | Start time | End time | Observation Hours per Day | Observation Hours per Year | Observation Hours per Solar Max. or Min. | Total of Observation Hours |
|---|---|---|---|---|---|---|---|---|---|---|
| **Solar Maximum** | 9.95000e-21 | 90.9 | 2013 | 27/06/2013 | 07h02m | 10h58m | 03h56m | 57h52m | 130h18m | 230h32m |
| | 1.01000e-20 | 90.9 | | 28/06/2013 | 07h02m | 12h58m | 05h56m | | | |
| | 1.19900e-20 | 94.6 | | 09/07/2013 | 00h00m | 00h00m | 24h00m | | | |
| | 1.17900e-20 | 94.6 | | 10/07/2013 | 00h00m | 00h00m | 24h00m | | | |
| | 9.91000e-21 | 112.6 | 2014 | 23/07/2014 | 00h00m | 09h26m | 09h26m | 09h26m | | |
| | 1.01000e-20 | 68.3 | 2015 | 15/07/2015 | 08h00m | 00h00m | 16h00m | 63h00m | | |
| | 9.96000e-21 | 68.3 | | 16/07/2015 | 00h00m | 00h00m | 24h00m | | | |
| | 9.74000e-21 | 68.3 | | 17/07/2015 | 00h00m | 23h00m | 23h00m | | | |
| **Solar Minimum** | 6.70000e-21 | 3.7 | 2019 | 18/06/2019 | 06h59m | 00h00m | 17h00m | 59h13m | 100h14m | |
| | 6.80000e-21 | 3.7 | | 19/06/2019 | 00h00m | 12h59m | 12h59m | | | |
| | 6.80000e-21 | 3.5 | | 04/07/2019 | 07h07m | 12h21m | 05h14m | | | |
| | 6.70000e-21 | 3.4 | | 20/08/2019 | 00h00m | 00h00m | 24h00m | | | |
| | 6.90000e-21 | 9.0 | 2020 | 06/07/2020 | 07h58m | 09h08m | 01h06m | 41h01m | | |
| | 6.80000e-21 | 9.0 | | 07/07/2020 | 00h00m | 11h59m | 11h59m | | | |
| | 6.70000e-21 | 9.0 | | 08/07/2020 | 00h00m | 11h59m | 11h59m | | | |
| | 6.90000e-21 | 9.0 | | 09/07/2020 | 00h00m | 11h58m | 11h58m | | | |
| | 6.90000e-21 | 9.0 | | 10/07/2020 | 08h00m | 11h59m | 03h59m | | | |

Moreover, Figure 2.7 from the "Handbook of physics and space environment" by Adolph S. Jursa (https://www.cnofs.org/Handbook_of_Geophysics_1985/Handbook.pdf ) shows that except for Lyman alpha, all solar UV shorter than 170 nm is absorbed above 100 km. There is not much EUV radiation left at our specific altitudes of interest (80 to 90/92 km).

Furthermore, the results from the study by Zhao et al., (https://agupubs.onlinelibrary.wiley.com/doi/10.1029/2020JD032418) indicate that despite the mesopause temperature's high sensitivity to solar radiation (as shown in Figure 6b of Zhao, 2020), the temperature has been decreasing over the 18 years of observation used in the study (ranging up to −0.14 K/year). (Also, in Figure 6a, the author specifies the corresponding error bars to the values shown in Figure 6b, which are large). Additionally, the mesosphere's altitude has also been decreasing over time at our latitude of interest (Figure 8a from Zhao, 2020). The study by Zhao et al. (2020) suggests that the declining mesopause reflects the contraction effect at lower altitudes caused by cooling from greenhouse gases, as discussed in other studies (Lübken et al., 2013; She et al., 2019; Yuan et al., 2019).

Considering these factors, the higher levels of EUV radiation alone cannot explain the higher mean altitude of PMSE layers during solar maximum. Further investigations, comparing the next solar maximum to the previous one, would be necessary. Additionally, it is important to note that again, this is a complex question with multiple parameters that vary with changing EUV radiation levels. This is discussed now in section 3.1 of the revised manuscript, accordingly to the reviewer's comment.

4- Abstract, lines 12-13. I don't see much evidence in your paper for gravity wave generation of the multilayers. In Figures 1 and 2, I can see evidence for a two multilayer form evolving into a monolayer. How can that be explained by gravity wavelength separation? From that description one would expect these multilayers to be separated equidistantly. My suggest is to point this out to the readership and stay with the observations. If you do have cases of say 3 or 4 layers, please add a figure

and show an example of these for the readership. Are they separated equidistantly? Some discussion would help. But from what I have seen so far, I would say that gravity waves are not the answer.

We are not attempting to provide a comprehensive explanation for the formation of multi layers using gravity waves. Instead, we mention a study conducted by (Li 2016), where they developed a model and varied the vertical wavelength of gravity waves. Interestingly, their findings indicated that as the number of layers increased, those layers tended to become thinner, with smaller wavelengths. In our own study, we observed a similar trend where an increase in the number of multi layers corresponded to thinner layers. As a result, we speculated that gravity waves might play a role in this phenomenon, although we did not state that they were a certain explanation. It is crucial to note that the study by (Li 2016) only modeled a single gravity wave, meaning there was only one wavelength considered. In reality, there are numerous gravity waves with varying wavelengths, making it more complex to determine the layer spacing (equidistant or not). Additionally, in (Li et al., 2016)'s study, the particle size was held constant while the vertical wavelength was varying, whereas in reality, we encounter a wide range of particle sizes. Therefore, the relationship between gravity waves and the formation of multi layers is not as straightforward.

5- Introduction. I have not seen any mention of metallic ions at ~92 km altitude which I believe come from the decomposition of micrometeoroids? Could these be the nuclei of water condensation? Can you add this to the discussion?

In our discussion, we mentioned that Meteoric Smoke Particles (MSPs) are the potential nucleation centers for ice particles. There are several other species discussed as possible nuclei for the ice particle formation and the work by Rapp and Thomas (2005) provides an overview. We are unaware of metallic ions serving as nucleation centers for ice particles. Based on the discussion by Rapp and Thomas, we speculate that metallic ions are possibly too small to start the nucleation process. Another limitation could be that the layers at 92 km would usually be well above the temperature minimum where frost temperatures are reached. Nevertheless, this is an interesting topic for future studies. The manuscript has been revised with more careful wording in accordance with the reviewer's comments.

"Meteor Smoke Particles (MSP), produced by meteor ablation and recondensation have been proposed as potential condensation nuclei along with several other potential nuclei (cg. Rapp and Thomas 2005). In addition to nucleation centers, the presence of cold temperatures and water vapor at mid and high latitudes at the mesopause during the summer months creates conditions favorable for ice particle formation Avaste (1993). Cold temperatures and water ice are known to be at the origin of another phenomenon called Noctilucent Clouds (NLC) Latteck et al. (2021)."

6- Line 19, your first sentence in the Introduction. A review paper on PMSE should be referenced here for the readership. Or if one does not exist, perhaps the discovery paper.

The manuscript has been updated accordingly to the reviewer's comment and the review paper on PMSE from Rapp and Lübken 2004 has been added as a reference in the introduction.

7- Line 46-47. I think perhaps you should mention solar EUV being stronger here during solar maximum? Of course in your data selection I do not know if you have chosen times when the ionosphere over Tromso was sunlit or not. Perhaps a mixture?

In response to the reviewer's question, the time period we selected for our data falls between June and August in Tromsø. During this time, daylight is almost always present. However, it is important to note that our study does not specifically focus on EUV. Therefore, detailing the amount of EUV during our chosen observation times is not what we want to focus on. Instead, we examine the level or background ionization, and we use the electron density at 92 km altitude as a proxy. Our main objective is not to determine the specific causes of ionization, such as particle precipitation or EUV (which goes beyond the scope of our study), but just to have a proxy for the ionization level in general. This point has been now clarified in the revised manuscript.

8- Line 86. Gravity wave breaking. If gravity waves break near the altitude of your study, that will go against the idea of the separation of the multilayers of PMSE by gravity wave wavelengths. So you will need to chose one route or the other. Gravity wave breaking needs a reference. I have talked to an

expert on this topic fairly recently and this person speculated that there was no such thing as gravity wave breaking. That the gravity waves evolved into multiple waves instead. So please be careful.

We agree that although it is believed that gravity waves will eventually break as the air becomes less dense at higher altitudes and the wave amplitudes increase; atmospheric dynamics are more complex, and we should not speculate on that in our paper. In order to comply with the reviewer's request, we now refer to a recent publication by Fritts et al. (2019) that presents new observations together with a description of the open questions in this area. (https://doi.org/10.1029/2019JD030298).

9- Line 102. Please give the location of the Magridal website. I do not see it in the acknowledgement section?

The link to the Madrigal website is in the "Data availability" section, instead of the "Acknowledgements" section.

10- Line 104. What is a "manda code"? Please explain to the readership.

Further information has been provided in the manuscript about the 'Manda' code, accordingly to the reviewer's comment:

"The EISCAT VHF radar utilizes many different experimental modes to collect data. The utilized pulse coding for the PMSE measurements we analyzed is referred to as 'Manda'. Detailed information regarding this coding can be found on the EISCAT website (https://eiscat.se/scientist/document/experiments/). For this study, we specifically analyzed data obtained using the 'Manda' code, because it is designed to detect low-altitude signals and layers in the mesosphere."

General Comment for the beginning of the paper. Much of this background material could be put into the Discussion and Conclusion Section. It would be good to shorten the front end of the paper and get to your analyses and results first. Then afterwards compare your results to theory and modeling.

Changes have been made accordingly to the reviewer's comment.

11- Lines 148. Please give references to Pearson and Spearman.

The references regarding the Pearson and the Spearman correlation coefficients have been added to the manuscript, accordingly to the reviewer's comment. Here is the updated version:

"In order to investigate different PMSE properties, we use the Pearson correlation coefficient and the Spearman's rank correlation coefficient to calculate the correlations between the different parameters, Wilks (1995), Myers and Well (2003)."

12- Figure 2. I have already made comments about Figure 2 (and Figure 1).

The reviewer's comments about this point were answered in question 4.

13- Figures 3, 4 and 5. The height distributions in all 3 of these histograms are quite broad. Compared to the small distances between the mean locations, I would draw the conclusion that there are no differences. Please discuss. You still can say that there are small differences but they may not be statistically significant. I doubt that the mean differences are statistically significant.

To answer the reviewer's comment about the statistical significance of the altitude values of the different layers, we calculated the p-values for all the combinations of all the layers. For our analysis, we used the most common significance level i.e., 0.05 (5%). The results are listed in Table A1 in the appendix. As the reviewer mentioned, some mean values can be very close to each other, indicating a p-value greater than 0.05, and some means are statistically different at 95% confidence, indicating a p-value less than or equal to 0.05. We now have computed p-values for all combinations of layers and parameters presented in our Figures (altitude distribution, electron density distribution, echo power distribution, and thickness of layers distribution). These p-values are listed in a

new Table A1. In the table below, we have included only the pertinent column, specifically the p-values related to the altitude distribution, to address the reviewer's comment.

| SOL MAX | P- value |
|---|---|
| Layers 1-2 | P = 0.6462 |
| Layers 1-3 | P < 0.0001 |
| Layers 1-4 | P = 0.0002 |
| Layers 2-3 | P < 0.0001 |
| Layers 2-4 | P = 0.0014 |
| Layers 3-4 | P = 0.8035 |
| SOL MIN | P- value |
| Layers 1-2 | P = 0.6808 |
| Layers 1-3 | P = 0.1098 |
| Layers 1-4 | P = 0.3030 |
| Layers 2-3 | P = 0.0481 |
| Layers 2-4 | P = 0.2284 |
| Layers 3-4 | P = 1.0000 |
| ALL LAYERS | P- value |
| Sol Max-Min | P < 0.0001 |

Additionally, we made some adjustments to the altitude distribution histograms (Figures 3, 4, and 5) by using smaller bins. The original EISCAT data we used have an altitude resolution of 360 m, so we chose this value as the new bin size on the x-axis, compared to the previous 1 km bins. Our focus is on the altitude range between 80 and 90 km. Within this range, we are examining the presence of PMSE multi layers. Given that we are describing up to 4 layers within this 10 km range and the altitude resolution of the data is 360 m, it can lead to the p-values often vary in terms of statistical significance. We selected the Manda code available in the EISCAT VHF radar data because it provides access to simultaneous measurements of the electron density above the PMSE, giving us insights into the general level of ionization. Also, the observations with Manda code give us the best resolution at altitudes of interest, out of the archived EISCAT VHF data. However, it could be valuable to explore the use of a different radars and/or codes with better resolution at PMSE altitudes in future studies of PMSE multi layers. Here are the new Figures 3, 4 and 5:

[Figure]

*Figure 3: Altitude distribution of the data for the **(a)** solar maximum and **(b)** solar minimum. Each subplot was its respective mean altitude represented with a red line on the graph, and specified in the legend together with one standard deviation.*

[Figure]

*Figure 4: Altitude distribution of the data during solar maximum for **(a)** mono layers, **(b)** multi layers with 2 layers, **(c)** multi layers with 3 layers, and **(d)** multi layers with 4 layers. Each subplot was its respective mean altitude represented with a red line on the graph, and specified in the legend together with one standard deviation.*

[Figure]

*Figure 5: Altitude distribution of the data during solar minimum for **(a)** mono layers, **(b)** multi layers with 2 layers, **(c)** multi layers with 3 layers, and **(d)** multi layers with 4 layers. Each subplot was its respective mean altitude represented with a red line on the graph, and specified in the legend together with one standard deviation.*

14- Page 5, 2 lines from the bottom. "Random forests"? Please describe for the readership.

We have added the following sentence, providing more detail to the reader about what random forests are:

"Random forests is a machine learning algorithm used for both classification and regression. In this algorithm, a number of decision trees are used during training phase to make predictions."

15- Lines 179-181. It is not known if the auroral particles precipitating are more energetic during solar maximum than during solar minimum at your magnetic latitude, so this argument is not valid. During solar maximum there are more magnetic storms than during solar minimum. But as mentioned previously these storm particles generally will precipitate at altitudes below that of Tromso. And the storm particles form the ring current and have energies of 10 to 300 keV. These particles will deposit their energy well below 92 km.

We have modified the manuscript accordingly to the reviewer's comment. We only stated our observations:

"The average altitude of all layers together is higher during solar maximum than during solar minimum (see Fig. 3)."

16- Line 209. Is the electron density decrease with number of layers statistically significant? This seems weak.

We have computed p-values for all combinations of layers and parameters shown in our Figures (altitude distribution, electron density distribution, echo power distribution, and thickness of layers distribution). These p-values can be found in a new Table A1. In the table below, we have included only the pertinent column, specifically the p-values related to the electron density, in order to address the reviewer's comment:

| SOL MAX | P- value |
|---|---|
| Layers 1-2 | P < 0.0001 |
| Layers 1-3 | P = 0.0003 |
| Layers 1-4 | P = 0.0831 |
| Layers 2-3 | P = 0.0804 |
| Layers 2-4 | P = 0.4000 |
| Layers 3-4 | P = 1.0000 |
| SOL MIN | P- value |
| Layers 1-2 | P < 0.0001 |
| Layers 1-3 | P < 0.0001 |
| Layers 1-4 | P < 0.0001 |
| Layers 2-3 | P < 0.0001 |
| Layers 2-4 | P = 0.0091 |
| Layers 3-4 | P = 0.5707 |
| ALL LAYERS | P- value |
| Sol Max-Min | P < 0.0001 |

17- Figure 12 and elsewhere (many other figures), the differences seem to be small and not statistically significant?

We have computed p-values for all combinations of layers and parameters shown in our Figures (altitude distribution, electron density distribution, echo power distribution, and thickness of layers distribution). These p-values can be found in Table A1. In the table below, we have included only the pertinent column, specifically the p-values related to the thickness of the layers (Figures 12, 13 and 14), to address the reviewer's comment:

| SOL MAX | P- value |
|---|---|
| Layers 1-2 | P < 0.0001 |
| Layers 1-3 | P < 0.0001 |
| Layers 1-4 | P < 0.0001 |
| Layers 2-3 | P < 0.0001 |
| Layers 2-4 | P < 0.0001 |
| Layers 3-4 | P = 0.0002 |
| SOL MIN | P- value |
| Layers 1-2 | P < 0.0001 |
| Layers 1-3 | P < 0.0001 |
| Layers 1-4 | P < 0.0001 |
| Layers 2-3 | P < 0.0001 |
| Layers 2-4 | P < 0.0001 |
| Layers 3-4 | P = 0.0002 |
| ALL LAYERS | P- value |
| Sol Max-Min | P < 0.0001 |

18- Conclusions, first paragraph. Yes you are correct, but I think the differences are not statistically significant.

We have computed p-values for all combinations of layers and parameters shown in our Figures (altitude distribution, electron density distribution, echo power distribution, and thickness of layers distribution). These p-values can be found in Table A1. Here is Table A1:

|  |  | Altitude | Electron Density | Echo Power | Thickness |
|---|---|---|---|---|---|
| Solar Maximum | Layers 1-2 | P = 0.6462 | P < 0.0001 | P < 0.0001 | P < 0.0001 |
|  | Layers 1-3 | P < 0.0001 | P = 0.0003 | P < 0.0001 | P < 0.0001 |
|  | Layers 1-4 | P = 0.0002 | P = 0.0831 | P < 0.0001 | P < 0.0001 |
|  | Layers 2-3 | P < 0.0001 | P = 0.0804 | P < 0.0001 | P < 0.0001 |
|  | Layers 2-4 | P = 0.0014 | P = 0.4000 | P < 0.0001 | P < 0.0001 |
|  | Layers 3-4 | P = 0.8035 | P = 1.0000 | P = 0.0012 | P = 0.0002 |
| Solar Minimum | Layers 1-2 | P = 0.6808 | P < 0.0001 | P = 0.3483 | P < 0.0001 |
|  | Layers 1-3 | P = 0.1098 | P < 0.0001 | P = 0.0009 | P < 0.0001 |
|  | Layers 1-4 | P = 0.3030 | P < 0.0001 | P = 0.0001 | P < 0.0001 |
|  | Layers 2-3 | P = 0.0481 | P < 0.0001 | P < 0.0001 | P < 0.0001 |
|  | Layers 2-4 | P = 0.2284 | P = 0.0091 | P < 0.0001 | P < 0.0001 |
|  | Layers 3-4 | P = 1.0000 | P = 0.5707 | P = 0.0728 | P = 0.0002 |
|  | Sol Max. - Min. | P < 0.0001 | P < 0.0001 | P < 0.0001 | P < 0.0001 |

Below is Table A2 where we derived the p-values for the correlation coefficients for all layers together during solar maximum and solar minimum:

| P-Value |  | Solar minimum | | | |
|---|---|---|---|---|---|
|  |  | Electron density | Echo power | Thickness | Altitude |
| Solar maximum | Electron density |  | 1,53E-27 | 1,38E-54 | 1,06E-04 |
|  | Echo power | 1,02E-203 |  | 0 | 2,51E-28 |
|  | Thickness | 0 | 0 |  | 1,94E-30 |
|  | Altitude | 0,772 | 2,24E-03 | 0,0175 |  |

Finally, below is Table A3 where we derived the p-values for the correlation coefficients for the mono and multi layers separately, during solar maximum and solar minimum:

| P-Value |  | Solar minimum | | | |
|---|---|---|---|---|---|
|  |  | Electron density | Echo power | Thickness | Altitude |

| | | | | |
|---|---|---|---|---|
| **Solar maximum** | Electon density | (grey) | Layer1 = 1,86E-19
Layer2 = 1,49E-14
Layer3 = 4,17E-04
Layer4 = 0,0489 | Layer1 = 1,02E-59
Layer2 = 2,84E-08
Layer3 = 4,17E-04
Layer4 = 0,00542 | Layer1 = 2,08E-27
Layer2 = 0,800
Layer3 = 0,165
Layer4 = 0,455 |
| | Echo power | Layer1 = 4,06E-183
Layer2 = 5,68E-58
Layer3 = 4,29E-12
Layer4 = 3,19E-05 | (grey) | Layer1 = 3,58E-139
Layer2 = 5,62E-112
Layer3 = 4,17E-04
Layer4 = 6,96E-22 | Layer1 = 0,0760
Layer2 = 2,30E-16
Layer3 = 2,51E-14
Layer4 = 5,92E-04 |
| | Thickness | Layer1 = 0
Layer2 = 9,23E-99
Layer3 = 1,65E-17
Layer4 = 3,60E-05 | Layer1 = 4,186E-319
Layer2 = 0
Layer3 = 8,51E-205
Layer4 = 1,89E-32 | (grey) | Layer1 = 2,87E-10
Layer2 = 8,82E-06
Layer3 = 4,17E-04
Layer4 = 0,0418 |
| | Altitude | Layer1 = 1,80E-07
Layer2 = 5,19E-04
Layer3 = 0,194
Layer4 = 0,305 | Layer1 = 6,87E-10
Layer2 = 5,85E-04
Layer3 = 5,32E-06
Layer4 = 6,02E-04 | Layer1 = 2,93E-19
Layer2 = 0,592
Layer3 = 0,0288
Layer4 = 0,174 | (grey) |

19- As stated earlier, it is best to stay with the observations. If they fit theory/code results, fine. If they do not, then fine again.

We have carefully rephrased the conclusion accordingly to the reviewer's comment, and adopted careful wording in the cases where the p-value did not allow us to draw a conclusion. Here is the only instance where this happened in the conclusion:

"Based on our investigation, we have found that the electron density at 92 km altitude and the echo power are positively correlated with the thickness for all the layers and for both solar maximum and solar minimum, except for four multi layers at solar minimum."

Also, we have included the following statement in the conclusion, referring to question 3 of the reviewer:

"While differences between the results from observations during solar maximum and during solar minimum considering all the layers together are statistically significant, the cause for the differences needs to be confirmed by future studies."

New reference:
Rapp, M., & Thomas, G. E. (2006). Modeling the microphysics of mesospheric ice particles: Assessment of current capabilities and basic sensitivities. *Journal of Atmospheric and Solar-Terrestrial Physics*, *68*(7), 715-744.

---

## Author Comment (AC2)

**Response to Reviewer 2:**

We thank the reviewer for the constructive and helpful comments, which helped us to improve the manuscript. We took all comments into account when revising the manuscript. In the text below we describe the modifications and list our responses together with the reviewer's comments that are repeated here in blue color. Our answers are given in black. In addition to the modifications listed below we make revisions to take into account the comments from Reviewer 1. We include a discussion of the statistical significance of the results. And we rephrase parts of the text to adjust the manuscript to the modifications made.

**General comments**

The article investigates properties of PMSE layers, more precisely of multiple layers, during periods of enhanced and minimum solar activity. The article is clearly structured. The literature review listed in the introduction covers the necessary background information on this topic. The data selection and analysis methods are presented and referenced in a very understandable way in section 2. The analysis of the data and discussion of the results is summarised in a structured way in one chapter. Overall, I see the article as a valuable contribution to the exploration of one aspect of the long-known but still complex phenomenon of PMSE. I would like to make a suggestion that I think could improve the readability of the paper and have listed some specific comments.

The study highlights the characteristics of PMSE in terms of their organisation into multi-layered structures, but the actual multi-layered structures are somewhat lost, at least in some illustrations. For example, the mean value of a distribution of parameters obtained from signals organised in multi-structures, as shown in Figures 4, 5, 10, 11, 13 and 14, says not much about the properties of the parameter with respect to the multi-structure. Rather, it represents the properties of a virtual layer that is organised into sub-layers. With the width of the layers considered further on, it then becomes complicated, as here the widths of the layers that occur simultaneously at different heights are combined. Therefore, I would recommend the authors to separate the distributions of the parameters in these figures for the multistructures found and to colour-code them, for example, and also to treat them separately in the analysis. Then, for example, in Fig.4b two distributions in two colours around two mean values would be shown in Fig.4c three distributions in three colours around three mean values and so on. With these separated parameters, detailed statements can be made about peak height, thickness, signal strength with regard to the occurrence in multilayer-structures and also in relation to the periods of solar maximum and minimum. This becomes particularly interesting and meaningful when, for example, the comparison to the NLC and the underlying mechanisms is made in chapter 3.4. Implementing this recommendation would, in my point of view, improve the readability of the article with regard to the multiple layers, because one would then see their distribution in combination with an improved bin resolution (see below) in the above-mentioned figures.

In the following section comprising three pairs of graphs, we address the reviewer's comments mentioned above, by providing and subsequently discussing the requested graphs. In each case, the solar maximum data is displayed at the top of the graph pair, while the solar minimum data is presented below it.

Altitude distribution, solar maximum:

[Figure]

Altitude distribution, solar minimum:

[Figure]

These graphs are quite intriguing, revealing new insights that were not apparent in the previous averaged representations. These figures are now present in the revised version of the manuscript (New Figures 4 and 5), and the manuscript's text has been changed accordingly, mentioning the new information and results that these graphs bring. Prior to the separation of layers within each set of multilayers, we encountered challenges in

achieving statistical significance, with some p-values exceeding 0.05. As a reminder, the p-values for the different altitude distribution graphs before layer separation are presented in the following Table:

| SOL MAX | P- value |
|---|---|
| Layers 1-2 | P = 0.6462 |
| Layers 1-3 | P < 0.0001 |
| Layers 1-4 | P = 0.0002 |
| Layers 2-3 | P < 0.0001 |
| Layers 2-4 | P = 0.0014 |
| Layers 3-4 | P = 0.8035 |
| SOL MIN | P- value |
| Layers 1-2 | P = 0.6808 |
| Layers 1-3 | P = 0.1098 |
| Layers 1-4 | P = 0.3030 |
| Layers 2-3 | P = 0.0481 |
| Layers 2-4 | P = 0.2284 |
| Layers 3-4 | P = 1.0000 |
| ALL LAYERS | P- value |
| Sol Max-Min | P < 0.0001 |

However, following the layer separation, it is noteworthy that almost all p-values (except for 1 case) associated with all the possible combinations of all the individual layers have now attained statistical significance, as shown in the following Table (This table is now present in the appendix of the revised manuscript):

| P-Values | | Solar Minimum | | | | | | | | | |
|---|---|---|---|---|---|---|---|---|---|---|---|
| | | Mono Layers | Layers 1 of 2 | Layers 2 of 2 | Layers 1 of 3 | Layers 2 of 3 | Layers 3 of 3 | Layers 1 of 4 | Layers 2 of 4 | Layers 3 of 4 | Layers 4 of 4 |
| Solar Maximum | Mono Layers | | P<0.0001 | P<0.0001 | P<0.0001 | 0.3618 | P<0.0001 | P<0.0001 | P<0.0001 | 0.0027 | P<0.0001 |
| | Layers 1 of 2 | P<0.0001 | | P<0.0001 | P<0.0001 | P<0.0001 | P<0.0001 | P<0.0001 | 0.0268 | P<0.0001 | P<0.0001 |
| | Layers 2 of 2 | P<0.0001 | P<0.0001 | | P<0.0001 | P<0.0001 | P<0.0001 | P<0.0001 | P<0.0001 | P<0.0001 | P<0.0001 |
| | Layers 1 of 3 | P<0.0001 | P<0.0001 | P<0.0001 | | P<0.0001 | P<0.0001 | 0.0106 | P<0.0001 | P<0.0001 | P<0.0001 |
| | Layers 2 of 3 | P<0.0001 | P<0.0001 | P<0.0001 | P<0.0001 | | P<0.0001 | P<0.0001 | P<0.0001 | P<0.0001 | P<0.0001 |
| | Layers 3 of 3 | P<0.0001 | P<0.0001 | 0.0002 | P<0.0001 | P<0.0001 | | P<0.0001 | P<0.0001 | P<0.0001 | 0.0001 |
| | Layers 1 of 4 | P<0.0001 | P<0.0001 | P<0.0001 | 0.0001 | P<0.0001 | P<0.0001 | | P<0.0001 | P<0.0001 | P<0.0001 |
| | Layers 2 of 4 | P<0.0001 | 0.0448 | P<0.0001 | P<0.0001 | P<0.0001 | P<0.0001 | P<0.0001 | | P<0.0001 | P<0.0001 |
| | Layers 3 of 4 | 0.0411 | P<0.0001 | P<0.0001 | P<0.0001 | P<0.0001 | P<0.0001 | P<0.0001 | P<0.0001 | | P<0.0001 |
| | Layers 4 of 4 | P<0.0001 | P<0.0001 | P<0.0001 | P<0.0001 | P<0.0001 | P<0.0001 | P<0.0001 | P<0.0001 | P<0.0001 | |

Echo power distribution, solar maximum:

[Figure]

Echo power distribution, solar minimum:

[Figure]

In the case of echo power distribution, it's notable that after layer separation, the graphs depicting echo power distribution present challenges in terms of readability. We observe slightly higher values during solar maximum compared to solar minimum, however this trend was already apparent in the previous averaged versions of the graphs. Additionally, it is possible to see that with increasing number of multilayers, the echo power tends to decrease. However, this insight was also discernible in the previous averaged version of the graphs. In summary, the graphs depicting separated layers do not introduce novel information. Consequently, we have chosen to include them exclusively in our response to the reviewer, but not into the revised version of the manuscript.

Thickness distribution, solar maximum:

[Figure]

Thickness distribution, solar minimum:

[Figure]

In the case of thickness distribution, after separating the different sets of multilayers into individual layers, the graphs depicting thickness distribution present challenges in terms of readability. Moreover, this did not bring novel insights, and for this reason, we have chosen to present these graphs only in our response to the reviewer, and not in the revised version of the manuscript. It is possible to see that as the number of multilayers increases within our 10 km altitude span, each individual layer tends to become thinner. However, it was already possible

to see this trend in the previous averaged versions of the graphs. Visual inspection seems to indicate that the lower layer seems to be consistently the thickest layer for solar minimum.

In the manuscript, we explicitly note that we have generated separate figures for both echo power distribution and thickness distribution when considering separated layers. However, upon careful consideration, we have opted not to include these figures in the final manuscript.

Below are some specific suggestions to the authors that I think could be included in the article:

**Specific comments**

- P1, L18: I would not say that the waveform is characteristic of PMSE, even though it occurs occasionally if not frequently, especially in the thin layers. The authors have modified the manuscript accordingly to the reviewer's comment, in the following way:

"PMSE are strong radar echoes that are linked to extremely cold temperatures, and their height and thickness varies over time, Rapp and Lübken (2004) ."

- P2, L27: Latteck et al. (2021) deals with PMSE and should not be used as a reference for NLC. The authors have modified the manuscript accordingly to the reviewer's comment:

The reference by Latteck et al. (2021) was replaced by the reference by Schäfer et al. (2020).

- P3, L49ff: I would suggest to move the sentence starting with "The mesopause ..." further up in this section e.g. after the references in L23. The authors have modified the manuscript accordingly to the reviewer's comment, in the following way:

"... The charged aerosols contain water ice, which requires the presence of low temperatures, sufficient water vapor, and nucleation centers to foster heterogeneous condensation, (Latteck et al., 2021), (Cho and Röttger, 1997), (Rapp and Lübken, 2004). The mesopause, which marks the boundary between the mesosphere and the thermosphere, is characterized by the lowest temperatures in the atmosphere. Such low temperatures at PMSE altitudes are conducive to ice formation, and PMSE are known to be influenced by ice formation through the slowing of diffusion processes ..."

Section 2.1 : I suggest to include this section into the introduction section and rewrite the introduction section since some parts as e.g. gravitiy wave breaking and turbulence is already mentioned there.

The authors have modified the manuscript accordingly to the reviewer's comment. We re-wrote the Introduction section incorporating the old Section 2.1 in it. As a consequence, we removed the previous section 2.1, "Theory behind the formation of PMSE" and rearranged accordingly the beginning of the section 2.

- P4, L104: I suggest writing "manda"-experiment instead of manda code and either giving a reference to a publication describing this experiment configuration in detail or summarising the most important experiment parameters here in the text. The authors have modified the manuscript accordingly to the reviewer's comment, in the following way:

"...The utilized pulse coding for the PMSE measurements we analyzed is referred to as 'Manda'. Some parameters of the EISCAT VHF radar using the 'Manda' experiment are listed in Table 2. Detailed information regarding this  experiment can be found on the EISCAT website (https://eiscat.se/scientist/document/experiments/). For this study, we specifically analyzed data obtained using the 'Manda'  experiment, because it is designed to detect low-altitude signals and layers in the mesosphere. We chose a time resolution of 60 seconds and a height resolution of 0.360 km ..."

**Table 2.** Some parameters of the EISCAT VHF radar, the source of data for this paper. More information about the EISCAT documentation and radar system parameters can be found at:
https://eiscat.se/scientist/document/experiments/

| EISCAT VHF parameters | |
|---|---|
| Frequency | 223.4 MHz |
| Wavelength | 1.34 m |
| Bragg scale | 0.67 m |
| Peak power | 1.2 MW |
| Transmitted pulse scheme | Manda v 4.0 |
| Interpulse period | 1.5 ms |
| Time resolution | 4.8 s |
| Range resolution | 360 m |
| Spectral resolution | 2.6 Hz |
| Antenna Elevation | 90 deg, zenith |

- Fig.3, 4 and 5: Why are the height or altitude distributions of PMSE detections shown in bins of 1km in these figures, when the experimental height resolution is 0.36m? The authors have modified the manuscript accordingly to the reviewer's comment. Old Figures 3, 4 and 5 might be now in the Appendix section, but they have been re-plotted using bins of 0.36 km. Here are the new Figure 3, Figure A1 and Figure A2:

Figure 3:

[Figure]

Figure A1:

[Figure]

FigureA2:

[Figure]

- Fig.4 and 5: What is the average altitude of a multilayer and what can be deduced from this value? If you observe the PMSE over many years, you will notice that the distance between e.g. double layers can cover a very large range, whereas the actual layers can be very narrow.

We have modified the figures to allow for a detailed examination of individual layers in the different sets of multilayers, with respect to altitude. The average altitude for each of these distinct layers is now presented in the updated Figures 4 and 5, with corresponding discussions in the text. (These new figures are shown in the first answer we made to the reviewer in this document.) Within the text, among various topics, we delve into the lower layer within a set of two PMSE multilayers. We provide specific details about its altitude and discuss its potential correspondence to altitudes observed in Noctilucent clouds.

- P9, L179ff: Here, the lower altitude of the PMSE and especially of the NLC should be discussed, which, as far as I know, is hardly subject to annual fluctuations. The increased energy input during the solar maximum at lower altitudes might therefore have no influence on the formation of PMSE at lower altitudes, as the other necessary conditions such as ice are no longer present above a certain altitude.

Please refer to our response to the previous comment, where we discussed the altitude of the lower layer within a set of two multilayers. We drew a parallel with the NLC altitude and provided some new comments in the manuscript about this. We revised the manuscript's text in which we now propose that these observations may be attributed to trends unrelated to solar cycle effects, emphasizing the need for further investigations.

- Section 3.2 : There is still some discussion missing here. What does the distribution of the electron density as well as its maximum and standard deviation say about the organisation of the PMSE in mono or multilayer?

The standard deviation is just the spread of values around the mean, and statistical significance is addressed with the p-values in Table B2. Visual inspection shows us that the standard deviation decreases with increasing number of layers, but it is difficult to determine the physical significance of the trend. The authors are not aware of a missing discussion, however here is a summary of the different points discussed in Section 3.2:

The discussion in this section points out to the observer that electron density is consistently lower in monolayers, both during solar maximum and minimum conditions. This observation suggests that higher electron densities may be a prerequisite for the formation of multilayers. The relevant passage from the manuscript is as follows:

"A plausible argument could be made that higher electron densities at ionospheric altitudes might be necessary to observe multi-layered PMSEs."

A new argument about the statistical significance of these results is added to the manuscript:

"However, it is important to bear in mind that this trend is weak and that some P-values corresponding to the different combinations of layers in Fig. 7 and Fig. 8 are greater than 0.05, as shown in Table B2."

In fact, the standard deviation provides insight into the dispersion of electron density values around their respective means. It's important to note that the standard deviation alone does not convey statistical significance. To address the issue of statistical significance, we have included a table in the appendix (Table B2) containing, among other, the p-values for all combinations of layers presented in Figures 6, 7, and 8. This table offers relevant information, if one wants to compare the mean values with each other, specifically.

Returning to the discussion of standard deviations, they reflect the spread of electron density values around the mean. In Figure 7, representing solar maximum conditions, larger standard deviations indicate a greater diversity of electron densities during this period. Visual inspection confirms a wider range of electron densities, particularly at higher values, compared to Figure 8, which represents solar minimum conditions. This observation suggests that higher electron densities are recorded during solar maximum phases.

- Fig.12, 13 and 14: Why are the distributions of PMSE thickness shown in bins of 1km in these figures, when the experimental height resolution is 0.36m?

The resolution is not in bins of 1 km, but already in bins of one data point, where one data point is equivalent to a distance of 360 m altitude. Consequently, the bins are already equivalent to 0.360 km. Here is the corresponding text mentioned in Section 3.4 that already specified that:

"Each data point or altitude channel corresponds to a distance of 360m."

However, the Figures have been changed accordingly to a comment the Reviewer made in the following Technical Corrections section. The Figures 12, 13 and 14 have been re-plotted using kilometers as a unit for the x axis and the legend instead of the number of data points. The text has also been modified in the manuscript. Please see the new Figures 12, 13 and 14 in the section below.

**Technical corrections**

All the following points have been implemented in the revised version of the manuscript :

- P1, L21: Remove (km).

"These echoes occur between 80 and 90 kilometers (km) altitude."

- P1, L23: The correct use of references should be checked throughout the text, e.g. the references here should be placed in brackets. See also at [P2, L26], [P2, L27], [P4, L83], [P4, L90], [P4, L103] We modified the citations accordingly to the reviewer's comment.
- P2, L30: Remove (PMSE).

"Multi-layered Polar Mesospheric Summer Echoes (PMSE) have been the focus of several investigations."

- P2, L39: Remove (NLC). The text has been revised following the incorporation of comments from the first reviewer, and consequently, this particular comment has been resolved

- P8, L167: I would not write layers here but detections, e.g. "average peak altitude of PMSE height distribution", as the plots in Fig.3 are probably not a distribution of predetermined layers.

"The average peak altitude of PMSE height distribution, considering all PMSE detections, The average altitude of all layers together is higher during solar maximum than during solar minimum (see Fig. 3) "

- P11, L119ff : Replace echo power by average echo power.

"Further, in Fig. 10, we observe that the average echo power decreases as the number of multi-layers increase for solar maximum and the individual layers considered."

- Fig. 12, 13, 14: I would suggest to use the correct thickness in m or km at the x-axes as well as in the text instead of altitude intervals.

The following Figures have been included in the manuscript and replaced the old Figures 12, 13 and 14: New Figure 12:

[Figure]

New Figure 13:

[Figure]

New Figure 14:

[Figure]

The x axis has been changed in Figures 12, 13 and 14. Instead of showing a number of data points, now the thickness is expressed in km. The values of the mean and standard deviation in the legend have also been converted into km. Here are the modifications in the text:

"As shown in Fig. 12, the average thickness of the layers is higher during solar maximum, with an average of 1.59 km , compared to solar minimum, where the average thickness is 1.32 km ."

"The highest average layer thickness is obtained during solar maximum for mono-layers with an average of 2.15 km , while the lowest average of 0.87 km  is obtained during solar minimum, for 4 multi-layers."

"Knowing that one altitude channel corresponds to 360m, 3 altitude channels or more indicate a PMSE thickness of at least 1.08 km . Our findings show that 54.64 percent of PMSE occurrences resulted in thick layers of 1.08 km  or more. "

- P13, L256: Remove nanometers and the brackets.

"In their first experiment, Li et al. (2016) fixed the particle size at 10 nm ..."

---

## Referee Report (RR1)

| | |
|---|---|
| Journal: | Anales Geophysicae |
| Manuscript Number: | egusphere-2023-977 |
| Article Title: | Polar Mesospheric Summer Echo (PMSE) Multilayer Properties During Solar Maximum and Solar Minimum |
| Authors: | D. Jozwicki et al. |

**Referee Report**

The article deals with the characteristics of PMSE layers, focussing on multi-layers during periods of enhanced and lower solar activity. It is clearly and concisely structured. In the introduction, a comprehensive literature review is given, providing relevant background information on the topic. Section 2 explains the methods of data selection and analysis in an easily understandable way. The results obtained are presented and discussed in detail in section 3 and then summarised in section 4.

After reviewing the paper for the second time, I can say that the authors have adequately taken into account the recommendations and suggested corrections that I made in the first review. In its current form, the manuscript has become much easier to read and has provided new information on the subject under investigation. I therefore consider the manuscript suitable for publication in its present form.

---

## Referee Report (RR2)

| | |
|---|---|
| Journal: | Anales Geophysicae |
| Manuscript Number: | egusphere-2023-977 |
| Article Title: | Polar Mesospheric Summer Echo (PMSE) Multilayer Properties During Solar Maximum and Solar Minimum |
| Authors: | D. Jozwicki et al. |

**Referee Report**

Based on my last review of this manuscript, I can only repeat that the authors have adequately considered the recommendations and suggested corrections that I made during the first review. In its current form, the manuscript is clearly readable and has provided new information on the topic under investigation. I therefore consider the manuscript in its current form to be suitable for publication.

---

## Author Response (AR2)

**Response to Reviewer 1 / Round 2 – Minor revisions**

We thank the reviewer for the constructive and helpful comments, which helped us to improve the manuscript. We took all comments into account when revising the manuscript. In the text below, we discuss the questions raised by the reviewer and describe the modifications made in the manuscript. We list our responses together with the reviewer's comments that are repeated here in blue color. Our answers are given in black.

Minor Comments:

- In the Introduction section, the authors mention that PMSE multilayers are related to Noctilucent Clouds. What is not mention is does the height of the Noctilucent Cloud layer vary with solar cycle? This should be discussed in the Conclusion section of the paper in connection with PMSE multilayer height variations.

- In the Conclusion section of this paper, the authors have not discussed the cause of the multilayers. The authors should mention existing theories of this for the readership. In fact it would be highly beneficial for the readership if the authors could comment on the possible explanations. They may wish to agree or disagree with the theories and even propose one of their own. Right now the paper is purely observational. It should be broadened to give physical understanding to the journal readership.

- Finally the authors mention that "future work should include further investigating the connections between multi layered PMSE formation and winds and gravity waves". Please mention to the readership if such phenomena can be directly measured or only modeled. Do you mean the Fritts method for gravity waves? Please cite. What about wind measurements? It should be noted that JGRSP, 121, 2016 doi:10.1002/2016JA022499 was written for a mechanism to understand Sudden Stratospheric Warmings (the Berlin Effect). Atmospheric scientists contrarily believe that the cause is gravity waves, but no gravity wave event observations have been correlated to a SSW event. Only models, which are a bit of a disappointment. Thus without observation there is no resolution. You need to explain to the readership that there can be some resolution. Please comment on this as well.

**Our answers to the Reviewer's comments:**

- In the Introduction section, the authors mention that PMSE multilayers are related to Noctilucent Clouds. What is not mention is does the height of the Noctilucent Cloud layer vary with solar cycle? This should be discussed in the Conclusion section of the paper in connection with PMSE multilayer height variations.

We appreciate the reviewer's valuable input on this intriguing question. As referenced in our manuscript, the study by Schäfer et al. (2020) highlights the altitude variability of NLC layers. However, their study did not focus on investigating the solar cycle effects on the NLC layers, and their research period spans the summers from 2011 to 2018. Nevertheless, we would like to draw attention to the recent work by Vellalassery et al. (2024) (https://www.mdpi.com/2073-4433/15/1/88), which addresses the variation of NLCs throughout the solar cycle. Their findings provide additional insights into this aspect of NLC behavior.

Vellalassery et al. (2024) found that the altitude of NLC exhibits a positive correlation with the solar cycle primarily as a result of temperature changes induced by the solar cycle. The way NLC properties respond to the solar cycle varies across different latitudes, with higher latitudes (69° N and 78° N) showing more pronounced and similar responses compared to mid-latitudes (58° N).

Among those changes, there is the decrease in altitude of NLC over time. Here is an extract from the Vellalassery et al. (2024) paper :

"We found solar cycle responses in the vertical distribution profiles of ice particle number, mean radius and NLC brightness. The solar cycle influence is present at all altitudes and peaks at the altitude of maximum NLC brightness. The magnitude of the ice particle radius and brightness response increases with time, mainly due to the increase of $H_2O$, while the downward shift of the profiles is due to atmospheric shrinking."

Our findings align with those results, as we observed a lower altitude of the PMSE during the solar minimum period (years 2019 and 2020) compared to the solar maximum phase (years 2013 to 2015). We included a short discussion in the conclusion where we cite the Vellalassery et al. (2024) paper.

- In the Conclusion section of this paper, the authors have not discussed the cause of the multilayers. The authors should mention existing theories of this for the readership. In fact it would be highly beneficial for the readership if the authors could comment on the possible explanations. They may wish to agree or disagree with the theories and even propose one of their own. Right now the paper is purely observational. It should be broadened to give physical understanding to the journal readership.

We appreciate the reviewer's insightful observation and the opportunity to further elaborate on this aspect. As mentioned in our manuscript, while the formation conditions of PMSE are well-understood, the precise mechanisms of PMSE multilayer formation remain uncertain. However, other authors hypothesized other mechanisms through which multi-layered PMSE can be formed. (e.g., Li et al., 2016; Hoffmann et al., 2005).

Li et al. (2016) introduced a model where they explored variations in the vertical wavelength of gravity waves to assess their impact on PMSE multilayers. They observed a decrease in the number of multilayers with increasing wavelength. Similarly, Hoffmann et al. (2005) employed a model simulation to examine the formation of PMSE multilayers and concluded that "the layering of PMSE can be explained by the layering of ice particles due to subsequent nucleation cycles in the vicinity of the mesopause and following growth and sedimentation." While these studies suggest a potential connection between PMSE multilayers and gravity waves, it's important to note the limitations inherent in these models.

Also, Li et al. (2016) explored the impact of particle size variation in their model and observed that larger ice particles lead to a quicker decrease in layer altitude and make layer formation more challenging. They also suggested that the multi-layer structure pattern is influenced by the vertical wavelength of the gravity wave, ice particle size, and wind velocity generated by the gravity wave. ("…we feel confident that the pattern of the multi-layer structure, at least

partially, depends on the vertical wavelength of the gravity wave, the ice particle size and the wind velocity caused by the gravity wave.")

Hoffmann et al. (2005) suggested that according to their model calculations, the layering of PMSE is expected to be more pronounced in the presence of long period gravity waves: "According to our model calculations the layering of PMSE should be particularly pronounced in the presence of long period gravity waves, i.e., waves that allow for the "correct" timing between nucleation, growth and sedimentation on the one hand and the phase propagation of the wave on the other (i.e., if the wave period is too short, the wave will only lead to an up- and downward- motion of the PMSE-layer because the microphysical processes are too slow to lead to significant changes of ice number densities and radii [see *Rapp et al., 2002*]."

In conclusion, our hypothesis on the formation of multi-layered PMSE is that gravity waves transport particles into regions of low temperature, and varying altitude. In these conditions, ice particles can form and grow. This process may impact the size of ice particles, which in turn could affect their spatial distribution via sedimentation, and potentially influencing the formation of multilayers. We added a few sentences in the conclusion section regarding this mechanism.

- Finally the authors mention that "future work should include further investigating the connections between multi layered PMSE formation and winds and gravity waves". Please mention to the readership if such phenomena can be directly measured or only modeled. Do you mean the Fritts method for gravity waves? Please cite. What about wind measurements?

We thank the reviewer for for raising this intriguing point. It is possible to measure gravity waves using the EISCAT radar as the recent paper from Günzkofer et. al. (2023) shows (https://doi.org/10.5194/angeo-41-409-2023). Then, utilizing the dissipative anelastic gravity wave dispersion relation, Günzkofer et. al. (2023) derive vertical wind profiles within the lower thermosphere.This is a promising avenue for further measuring of gravity waves during PMSE occurrences. We included a comment in the conclusion where we cite the Günzkofer et. al. (2023) paper.

- It should be noted that JGRSP, 121, 2016 doi:10.1002/2016JA022499 was written for a mechanism to understand Sudden Stratospheric Warmings (the Berlin Effect). Atmospheric scientists contrarily believe that the cause is gravity waves, but no gravity wave event observations have been correlated to a SSW event. Only models, which are a bit of a disappointment. Thus without observation there is no resolution. You need to explain to the readership that there can be some resolution. Please comment on this as well.

While this topic is intriguing, it falls beyond the scope of this manuscript.

---

## Author Response (AR3)

**Response to the Editor:**

We thank the editor for the constructive and helpful comments, which helped us to improve the manuscript. We took all comments into account when revising the manuscript. In the text below we describe the modifications and list our responses together with the reviewer's comments that are repeated here in blue color. Our answers are given in black.

Comment from the Editor:

With the next file upload request, please update the "Competing interests" as follows: At least one of the (co-)authors is a member of the editorial board of Annales Geophysicae

Thank you for reminding us of this point, here is the updated version on the Competing interests section :

"At least one of the (co-)authors is a member of the editorial board of Annales Geophysicae. The peer-review procedure was conducted by an impartial editor, and the authors declare that they have no additional competing interests. The funders had no role in the design of the study; in the collection, analyses, or interpretation of data; in the writing of the manuscript, or in the decision to publish the results."